# Bioinspired "cage traps" for closed-loop lead management of perovskite solar cells under real-world contamination assessment

Huaiqing Luo[1,3], Pengwei Li[1,3], Junjie Ma [1] ✉, Xue Li[1], He Zhu[1], Yajie Cheng[1], Qin Li[1], Qun Xu [1], Yiqiang Zhang[1] ✉ & Yanlin Song [2] ✉

Despite the remarkable progress made in perovskite solar cells, great concerns regarding potential Pb contamination risk and environmental vulnerability risks associated with perovskite solar cells pose a significant obstacle to their real-world commercialization. In this study, we took inspiration from the ensnaring prey behavior of spiders and chemical components in spider web to strategically implant a multifunctional mesoporous amino-grafted-carbon net into perovskite solar cells, creating a biomimetic cage traps that could effectively mitigate Pb leakage and shield the external invasion under extreme weather conditions. The synergistic Pb capturing mechanism in terms of chemical chelation and physical adsorption is in-depth explored. Additionally, the Pb contamination assessment of end-of-life perovskite solar cells in the real-world ecosystem, including Yellow River water and soil, is proposed. The sustainable closed-loop Pb management process is also successfully established involving four critical steps: Pb precipitation, Pb adsorption, Pb desorption, and Pb recycling. Our findings provide inspiring insights for promoting green and sustainable industrialization of perovskite solar cells.

With the increasing global energy crisis, perovskite solar cells (PSCs) have emerged as a promising renewable energy technology for reducing the carbon footprint worldwide. PSCs offer superior compatibility and scalable fabrication, positioning them to revolutionize the photovoltaic market. However, significant challenges remain to be addressed before they can be practically applied. The volatile organic cations (such as $CH_3NH_3^+$ or $HNCH(NH_3)^+$) and soft lattice properties of PSCs are the sources of intrinsic instability, resulting in decreased device efficiency and long-term operational stability issues[1–3]. Notably, the decomposition of perovskite films can lead to the formation of toxic Pb-based compounds, including $PbI_2$, Pb, or PbO, when exposed to external stimuli such as moisture, illumination, and heat. These compounds can potentially leak into the ecosystem, raising environmental sustainability concerns[4,5].

Recent efforts focused on encapsulating PSCs to reduce Pb leakage. Encapsulation materials such as polyurethane, polyisobutylene, graphene and $Al_2O_3$ have been applied using physical hot-press or atomic layer deposition method[6–9]. However, these encapsulation layers cannot prevent Pb component diffusion if the device is broken under external stress due to the limited Pb capture capability. While the complex preparation process also increases the production costs. Chemical adsorption strategies employing functional chemical materials for external adsorption to minimize Pb leakage have been proposed. Most published works focus on developing semi-transparent Pb adsorbent that can be installed on the light-receiving side of PSCs. High optical transparency is required to avoid decreased photovoltaic performance, limiting the thickness of semi-transparent Pb adsorbent and compromising Pb adsorption capacity and ambient resistance. In

[1]Henan Institute of Advanced Technology, College of Chemistry, Zhengzhou University, Zhengzhou 450001, PR China. [2]Key Laboratory of Green Printing, Institute of Chemistry, Chinese Academy of Sciences, Beijing Engineering Research Center of Nanomaterials for Green Printing Technology, National Laboratory for Molecular Sciences, 100190 Beijing, PR China. [3]These authors contributed equally: Huaiqing Luo, Pengwei Li. ✉e-mail: junjiema@zzu.edu.cn; yqzhang@zzu.edu.cn; ylsong@iccas.ac.cn

practice, the leaked Pb component tends to flow towards the backside of the device due to gravity, and corresponding materials or strategies remain limited. Li et al. integrated a polymer mixture based on cation exchange resin (CER) and ultraviolet (UV) resin into the PSC[10]. The Pb component was adsorbed via the rapid cation exchange reaction between abundant sulfonic acid groups ($SO_3^-$) and $Pb^{2+}$, achieving 90% Pb sequestration efficiency. Nevertheless, the potential risk of secondary pollution from hazardous solid CER waste emerged as a environmental issue. Additionally, concerns regarding sustainable disposal strategies for end-of-life perovskite photovoltaic modules persist, owing to the potential threat of toxic Pb ions to the ecosystem and human health safety[11-14]. These drawbacks may seariouly hinder the commercial application of PSCs.

Conventional Pb leakage assessment are currently conducted under laboratory conditions, which do not accurately reflect the objective impact on the environment. These manually simplified experimental conditions would inevitably eliminate factors such as complex organic composition, heavy metals, and microorganisms[15,16]. Furthermore, persistent environmental pollution issues resulting from the end-of-life PSCs haven't been thoroughly investigated in real-word scenarios. Spider web possesses unique mesh structure and strong adhesion properties, making it an excellent biological material with exceptional hunting ability. Its natural protein molecular chain structure containing glycine ($NH_2$-$CH_2$-COOH), alanine ($NH_2$-CH[$CH_3$]-COOH), and serine ($NH_2$-CH[$CH_2$OH]-COOH), allowing for strong coordination with the target surface[17,18]. Inspired by these natural chemical components, we designed a functional Pb capture molecular structure (CONH-R-$NH_2$) based on ethylenediamine (EDA) and implanted it into mesoporous matrices (MM) to synthesize an eco-friendly and low-cost bioinspired cage traps (BCT). The BCT is capable of capturing leaked Pb ions from damaged devices, isolating against undesired moisture erosion and recycling Pb from end-of-life devices (Supplementary Table 1). The MM acts as reaction chambers that provide strong physical adsorption processes launched by capillary adsorption effects. This approach achieves synergistic Pb capture effect based on physical and chemical functions. Meanwhile, mathematical modeling was used to elaborate on Pb sorption kinetics based on Langmuir isothermal adsorption equations, where chemisorption induced by strong chelation dominates the Pb adsorption rate. By employing this inexpensive BCT, dissociative Pb ions can be effectively adsorbed up to 99.25%, achieving Pb sequestration efficiency exceeding 99% under extreme weather conditions. More importantly, rigorous and reliable testing procedures are demonstrated by placing PSCs in realistic environments such as Yellow River water and Yellow River soil, providing a benchmark for establishing a systematic and repeatable Pb leakage real-world assessment standard. Moreover, the PSCs maintain 92% of their initial efficiency at 25 °C and 50% RH for 1000 h. Sustainable closed-loop Pb management practices have been successfully established, including Pb precipitation, Pb adsorption, Pb desorption and Pb recycling. It is encouraging that the purity of recycled $PbI_2$ is comparable to commercial 99.99% $PbI_2$, and it does not compromise the photovoltaic performance of PSCs. This work addresses the issues of capturing leaked Pb ions from in-service PSCs and recycling Pb from end-of-life PSCs simultaneously to alleviate Pb contamination risk, as well as proposing reliable and systematic procedures for assessing hazardous Pb leakage in actual environmental scenarios. The proposed strategy will pave the way for accelerating perovskite photovoltaic technology toward sustainability, eco-friendliness and industrialization.

## Results

### Chemical synthesis design of the BCT
Spiders are known to ensnare their prey by spinning a delicate and adhesive web, where the viscous glycoproteins on the surface of the spider's web behave as viscoelastic solids that can deform like an ideal elastic rubber band. It is essential for retaining the insects trapped in the web long enough to be subdued by the spider[19], as shown in Fig. 1a. The adhesive properties of the spider mucilage arise from unique chemical components, including $NH_2$-$CH_2$-COOH, $NH_2$-CH[$CH_3$]-COOH and $NH_2$-CH[$CH_2$OH]-COOH. The characteristic amino (-$NH_2$) and carboxyl (-COOH) groups play a crucial role in the strength and viscosity properties due to strong hydrogen bonds, electrostatic attraction and van der Waals force, as shown in Supplementary Fig. 1(I),[20,21]. Meanwhile, the $NH_2$-terminal and COOH-terminal of non-repetitive regions at both ends of spider silk protein control the formation of spider silk fibers and polypeptide chain micelles adhesive through the protein secondary structural crosslinking induced by dehydration and condensation, as shown in Supplementary Fig. 1(II)[22]. Besides, the -COOH and -$NH_2$ groups provide selective complexation with metal ions via strong chemical coordination. Moreover, the unique radial and porous structure of spider web offers huge hunting space and flexibility (Supplementary Fig. 1(III)). Inspired by these unique features of spiders hunting, we designed a multifunctional mesoporous amino-grafted-carbon net (BCT) to capture dissociative divalent Pb ions and offer a steric shield effect under ambient stimulus (Fig. 1a and Supplementary Fig. 1). The MM, analogous to a spider's web, serves as reaction chambers, where the dissociative Pb ions can be confined within mesoporous cage space. -COOH groups are preliminarily grafted on the MM by hydrochloric acid activation method to form primary chemical reaction sites. Notably, EDA molecules containing two -$NH_2$ groups were self-assembled onto the activated MM to reinforce the Pb capture capability (analogous to viscous microemulsion on spider's web) (Supplementary Fig. 2).

As shown in Fig. 1b, the Fourier transform infrared spectroscopy (FTIR) of the MM shows characteristic peaks at around 3446 $cm^{-1}$ and 1630 $cm^{-1}$ assigned to the stretching vibration of the hydroxyl group (-OH) and carbonyl group (-C=O) of the -COOH group respectively, confirming the presence of -COOH in the MM induced by the hydrochloric acid activation treatment. Compared to MM, the BCT demonstrates new peaks regarding -N-H tensile vibration at around 3417 $cm^{-1}$ and the out-of-plane deformation vibration absorption of -$NH_2$ at around 880 $cm^{-1}$, indicating the terminal -$NH_2$ group from EDA molecular has been successfully grafted onto the BCT. Additionally, the strong vibration absorption peaks at 1633 $cm^{-1}$ and 1579 $cm^{-1}$ are related to the stretching vibration of the -C=O and deformation vibration of N-H of the acylamide group (-CONH), which indicates that p-π conjugation between the -C=O and the lone-pair electron of nitrogen atoms on amide groups has occurred[23]. Compared with the peak at 1630 $cm^{-1}$ belonging to -C=O group in -COOH of MM, the peak of the -C=O group in BCT shifts to 1633 $cm^{-1}$, demonstrating that the charge distribution around -C=O has changed and -COOH group has participated in the self-assembly grafting process of EDA. The strong peak at 1099 $cm^{-1}$ is attributed to the C-N antisymmetrical stretching vibration. Moreover, the X-ray photoelectron spectroscopy (XPS) analysis indicates the presence of substate signals of $N1s$ at 395.8 eV and 396.9 eV, proving that the -$NH_2$ and CONH-based groups exist on the BCT surface, as shown in Fig. 1c[24]. Given the spontaneous dehydration condensation reaction between the -$NH_2$ and -COOH terminal for amidating polypeptide chain in spider silk protein, it is confirmed that the EDA molecules are grafted onto the BCT surface. This reconstruction of surface states produces functional groups (-CONH-R-$NH_2$) via the dehydration and condensation between the -COOH and -$NH_2$ groups, triggered by the assistance of ultrasonic waves[25]. Based on the analysis of the molecular structure evolution, the chemical synthesis route of the BCT can be deduced as Eq. (1).

$$-COOH + NH_2 - C_2H_4 - NH_2 \rightarrow -CO - NH - C_2H_4 - NH_2 + H_2O \quad (1)$$

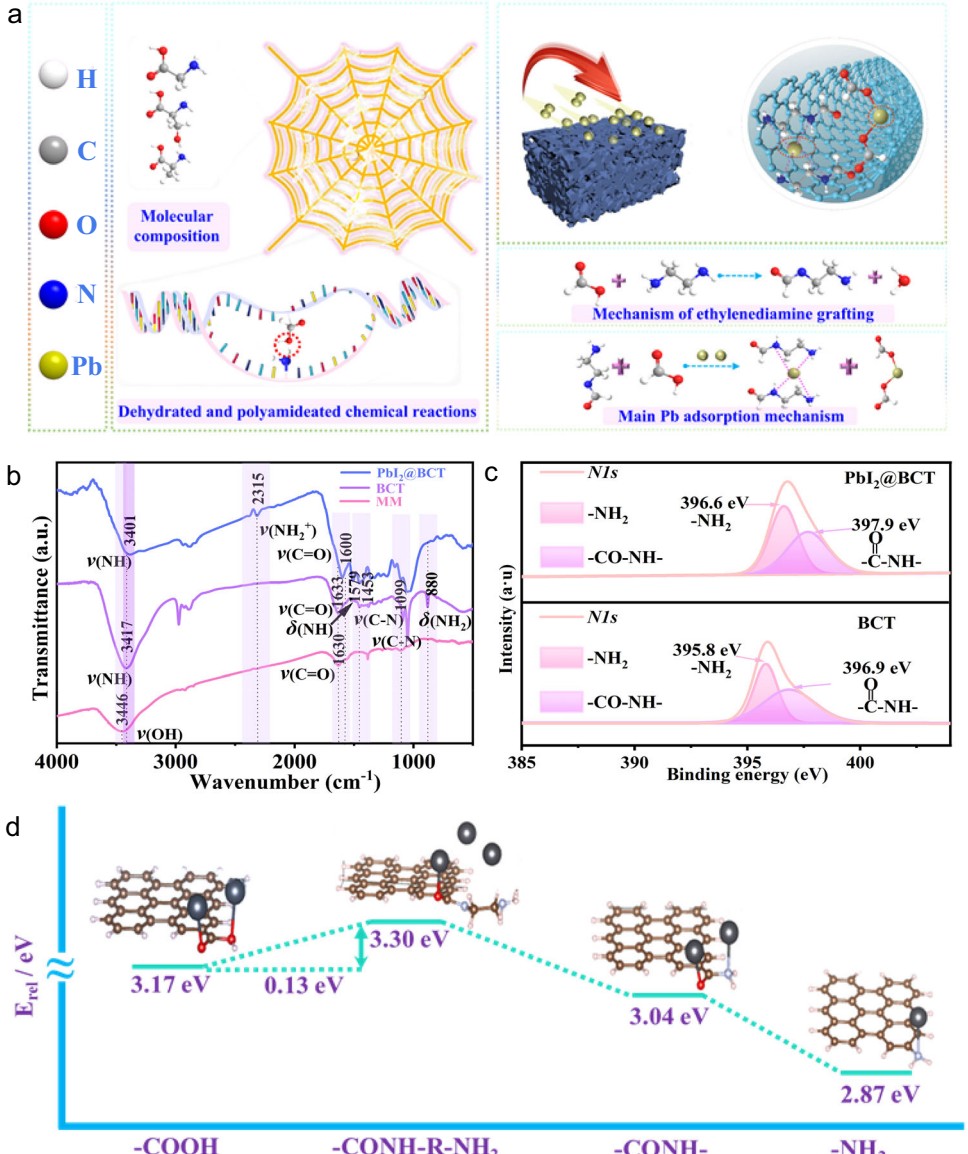

**Fig. 1 | Chemical synthesis design and chemical Pb capturing mechanism of the BCT. a** Hunting mechanism of spider web from molecular composition to overall structure (left) and Pb chelation behavior of BCT (right). **b** FTIR spectra of MM, BCT and PbI₂@BCT. **c** XPS spectra of *N1s* for MM, BCT and PbI₂@BCT. **d** Graphical representations of the computed adsorption energy model structures and results for -COOH, -CONH, -NH₂ and -CONH-R-NH₂.

## Pb capturing mechanism based on the chemical chelation

In order to understand the Pb capturing mechanism based on the chemical chelation between the functional groups that grafted on the BCT surface and Pb ions, the deviation of the signal peaks in XPS and FTIR are analyzed. Fig. 1c and Supplementary Fig. 3 show the XPS peak spectra of N and Pb elements of pure PbI₂ and PbI₂ mixed with BCT powders (referred as PbI₂@BCT samples). The peaks corresponding to Pb *4f5/2* and Pb *4f7/2* shift from 139.6 eV and 134.7 eV to higher binding energies of 140.6 eV and 135.8 eV for PbI₂@BCT samples, respectively. This indicates that the inner shell electrons of Pb ions possess enhanced binding energy due to the bonding state. Furthermore, the substate N signals in -NH₂ peak of 395.8 eV and the -CONH peak of 396.9 eV on the BCT surface are remarkably upward to 396.6 eV and 397.9 eV upon encountering Pb ions. Besides, in the Fig. 1b, the peak positions of -NH and -C=O in the -CONH group of PbI₂@BCT samples obviously downward shift from 3417 cm⁻¹ and 1630 cm⁻¹ to 3401 cm⁻¹ and 1600 cm⁻¹ in FIIR spectra, respectively. These results provide strong evidence regarding the chemical chelation between the -CONH and Pb ion. -NH and -C=O as Lewis base adducts in -CONH group with

electron-donating properties offer delocalized electrons, thus altering the surrounding chemical environment and inducing peripheral coulomb interactions with positive Pb ions[26]. Interestingly, out-of-plane deformation vibration absorption peak at 880 cm⁻¹, assigning to the terminal -NH₂ group, disappears while accompanied by appearing new telescopic vibration peak of -NH₂⁺ at 2315 cm⁻¹. This is due to the delocalized lone pair of electrons on the nitrogen atom shifting to the positively charged Pb²⁺ to form metal coordination complex[27].

To understand how the molecular configuration influences the chelating strength, first-principles density-functional theory (DFT) was conducted to calculate binding energies between different chemical groups and Pb ions. As depicted in Fig. 1d, compared with the original -COOH (3.17 eV) and -NH₂ (2.87 eV) groups, -CONH as one of the reaction products shows a binding energy of 3.04 eV. This is because the -C=O and -NH in the -CONH exhibit Lewis basicity where lone pair electrons in oxygen and nitrogen atoms delocalized to form the basis of the strong interaction[26,28]. Interestingly, -CONH-R-NH₂ exhibits supreme binding energies of 3.3 eV, due to the coupling molecular configuration containing multiple Pb binding sites such as -NH₂ and

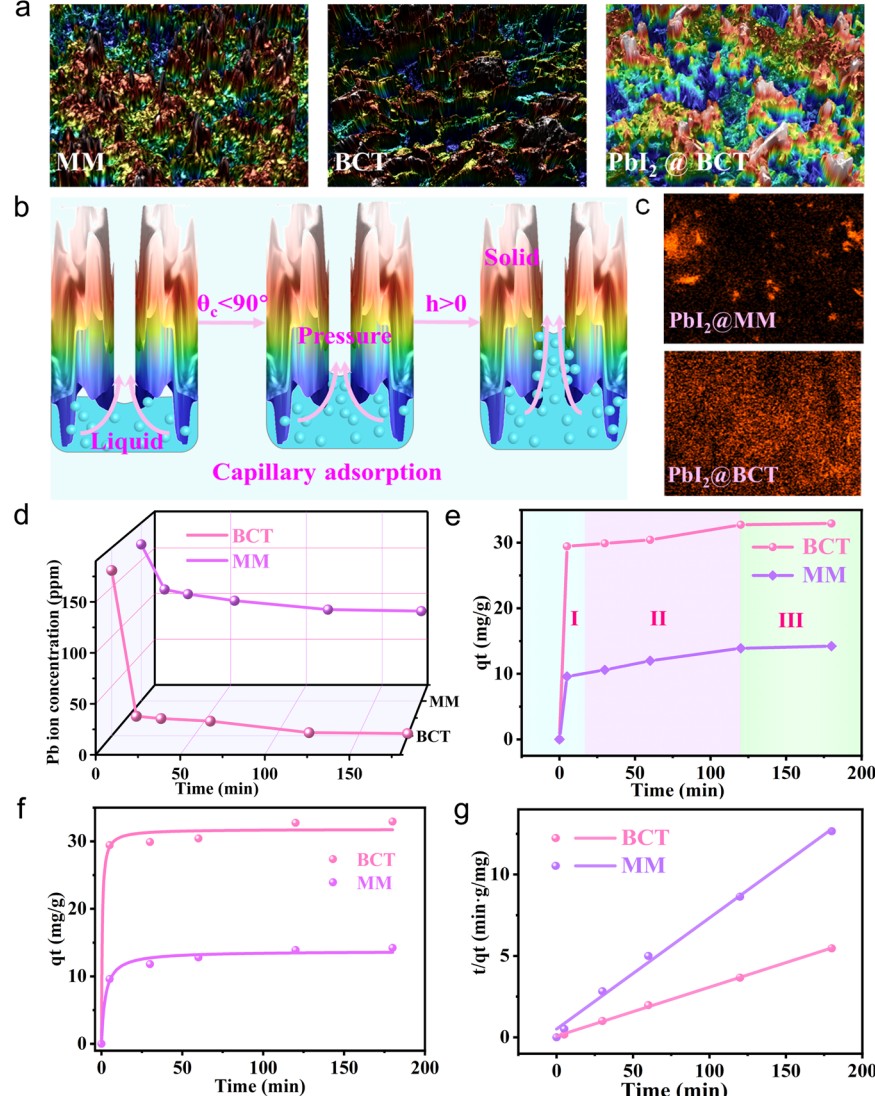

**Fig. 2 | Physical Pb adsorption mechanism and the mathematical modeling for sorption kinetics of BCT. a** 3D optical profilometer measurement on the MM, BCT and PbI₂@BCT surfaces. **b** Schematic of capillary adsorption. **c** EDS results towards Pb elemental after adsorption of PbI₂ by MM and BCT (The Pb content in MM and BCT is 0.1 At% and 0.4 At% respectively.). **d** The Pb ion concentration change during the process of MM and BCT reacting with PbI₂. **e** Process analysis of MM and BCT reacting with PbI₂ (stage I:0–5 min, stage II:5–120 min and stage III:120–180 min). **f** Pseudo-second-order adsorption kinetic process of Pb ions by MM and BCT at the same temperature. **g** The pseudo-second-order fitting result ($K_{2\text{-}BCT}$ and $K_{2\text{-}MM}$ are further deduced to be 0.0690 and 0.0306 g·(mg/min)).

-CONH, achieving synergistic adsorption of Pb ions (Supplementary Fig. 4). According to the results, it can be inferred that a significant chemical interaction exists between the BCT group and PbI₂. The chemical adsorption capacity of BCT for Pb ions has been greatly improved via the step-by-step process of hydrochloric acid activation and EDA molecular self-assembly.

The macroscopic morphology of the MM surface is observed to change in a large-scale coupled with the EDA self-assembly grafting process, which is confirmed by scanning electron microscopy (SEM), three-dimensional optical profilometer (3DOP), and high-pressure chemisorption. The synthesis of BCT can be divided into three stages, including (I) Initial state, (II) Ultrasound-assisted structural separation stage and (III) Surface chemical reconfiguration stage (Supplementary Fig. 5a(I)). The surface morphology of BCT in these three different stages was observed via SEM to figure out what stage and by what reaction the number of pores of MM increases. As shown in Supplementary Fig. 5a(II) and Supplementary Fig. 5b, in the stage I, the pristine MM shows a relatively smooth surface and the number of pores was estimated around 13 per 100 μm². In the stage II, MM was treated

by high-frequency ultrasonic vibration over 90 min to reduce the particle size, providing better dispersion in solution. It can be clearly seen that the adjacent pores are fused together and new pores appear, with an estimated number of pores around 36 per 100 μm² and the larger average pore diameter (Supplementary Fig. 5b). In stage III, EDA molecular is grafted on the MM to reconstruct the surface chemical state with the assistance of high-frequency ultrasonic vibration, producing the CONH-R-NH₂ molecular configuration. The synthesized BCT surface presents a rougher surface with more pores appearing (Supplementary Fig. 5a). The number of the pores is around 52 per 100 μm². The external surface area of BCT is quantitatively evaluated as 108.4398 m²g⁻¹ via Brunauer–Emmett–Teller surface area and pore size analyzer, which is more than twice as large as MM (Supplementary Fig. 6). The detailed open cavities of BCT were further visualized by 3DOP, as shown in Fig. 2a and Supplementary Fig. 7. The average surface roughness of BCT increases from 22.393 to 32.535 μm after grafting EDA due to generating a large number of new pores. It is reasoned that the high-frequency ultrasonic vibration during the EDA self-assembly grafting process physically reconstructs the surface

structure to some extent due to the potential material plasticity. The porous media is endowed with elongated inner cavities that can be regarded as capillaries for triggering the capillary adsorption action[25]. Driven by liquid surface tension, the immersed liquid can be spontaneously adsorbed by inner cavities and rise along the contact interface, as shown in Fig. 2b. The height (h) of the raised liquid level can be calculated by Eq. (2)[29].

$$(\gamma_{SG} - \gamma_{SL})2\pi a \approx \rho g \pi a^2 \qquad (2)$$

It can be concluded that the h of the raised liquid level in pipe diameter is proportional to the contact area between liquid and MM or BCT. Combined with Young's Equation: $\cos\theta_c = \frac{\gamma_{SG}-\gamma_{SL}}{\gamma_{LG}}$, h is further defined as[30]:

$$h \approx 2\frac{L_c^2}{a}\cos\theta_c \qquad (3)$$

Where, $\gamma_{SG}$, $\gamma_{SL}$, $\gamma_{LG}$ are the surface tension coefficients between solid-gas and solid-liquid, liquid-gas, respectively; $a$ is the capillary radius; $\rho$ is the liquid density; $g$ is the acceleration of gravity; $h$ is the height of the liquid level rise; $\theta_c$ is the contact angle; $L_C = \sqrt{\frac{\gamma_{LG}}{\rho g}}$.

It can be seen that the theoretical result of liquid level raised height is $h > 0$ when the contact angle between the solution and the adsorbate is an acute. The smaller the contact angle, the larger the h value, indicating that capillary adsorption is more likely to occur. Therefore, we further verified the raised liquid level of MM and BCT via testing the wettability of the deionized water (DI water) on the MM and BCT surfaces. As shown in Supplementary Fig. 8, BCT exhibits better wettability with a lower contact angle (38.5°) compared with the MM with 58.9°contact angle, indicating the BCT is more conducive for capillary adsorption. Therefore, a synergistic capture effect based on strong physical adsorption and chemical chelation can be realized.

## Pb capturing capacity analysis

The concentration of residual $PbI_2$ concentration can be qualitatively characterized by UV-visible absorption spectra and the amount of $PbSO_4$ precipitation. As shown in Supplementary Figs. 9 and 10, the BCT sample synthesized with 7 mL EDA produced the minimum $PbSO_4$ precipitation, which also supported by the lowest UV absorbance of residual $PbI_2$ solution (the sample as optimized target group hereafter in the following section).

To further simulate the Pb leakage from the degraded perovskite film, $FAPbI_3$ perovskite powders were immersed in DI water, where the Pb ions testing paper was used to quickly detect the Pb concentration and check the Pb capturing capacity of BCT and MM in the contaminated water, respectively (Supplementary Fig. 11). The Pb ions concentration in the original solution exceeded 100 ppm, confirming the degraded perovskites would induce severe Pb diffusion in the aqueous solution. Distinctly, the Pb ions extracted from the BCT-perovskite solution are close to 0 ppm, which is much lower than the MM-perovskite solution with a level of 20 ppm. The resultant MM and BCT adsorbed with Pb ions were further in-situ characterized by X-ray energy spectrum analysis (EDS) to detect the Pb element distribution. As shown in Fig. 2c, the Pb content in BCT is 0.4 At%, which is higher than that in MM (0.1 At%). The Pb ions adsorbed on the material surface also alter the roughness, which is depicted in Fig. 2a. The average surface roughness of BCT decreases by 18.115 μm after adsorbing $PbI_2$ compared with MM. These results strongly prove that BCT exhibits better Pb capturing capacity than MM.

The dynamics evolution of Pb adsorption concentration which quantitatively determined via an inductively coupled plasma emission spectrometer (ICP-OES) under different duration time is summarized

in Fig. 2d. MM adsorbed 42.57% of the Pb in the solution after 180 min. In contrast to MM, BCT gives a much faster adsorption capacity that 88.24% of the Pb ions are captured just within 5 min. Moreover, over 98.60% of Pb ions can be adsorbed in an aqueous solution by BCT in the final dynamic equilibrium stage. To further reveal the adsorption reaction process, mathematical models are established based on Langmuir isothermal adsorption equation. The distribution coefficient ($K_d$) was calculated to explore the relative ratio of Pb concentration in the solid phase (MM, BCT) and liquid phase when adsorption reaches equilibrium (see Supplementary Note 1 for detailed calculation). The $K_d$ value of BCT is as high as 14134.76 mL g$^{-1}$, which is nearly two orders of magnitude higher than that of MM (148.27 mL g$^{-1}$). This suggests that BCT as sorbent possesses a better affinity for Pb ions and thus offers much stronger Pb capturing capacity[31]. The whole adsorption curves of BCT and MM can be further divided as three adsorption reaction processes including (I) chemical dominated adsorption stage (0–5 min), (II) physical dominated adsorption stage (5–120 min) and (III) equilibrium adsorption stage (120–180 min), as shown in Fig. 2e. In stage I, the Pb ion concentration gradient between the liquid and solid phases promotes the rapid diffusion of Pb ion toward the adsorbent surface. The calculation processes, including adsorption capacity and pseudo-second-order kinetics, are summarized in Supplementary Note 2.

Interestingly, BCT has three times higher Pb adsorption capacity than MM in stage I, illustrating the critical role of sufficient chemical adsorption sites on the surface to immobilize dissociative Pb ions in the initial state. The adsorption process transitions into the physical-dominated stage (stage II) when the chemical active sites on the adsorbent surface become saturated. The adsorption rate slows down, indicating that Pb ions are slowly diffusing into the pore channels of the adsorbent through capillary adsorption mechanism. In stage I, the adsorption capacity of Pb by MM and the BCT was 9.60 and 29.44 mg/g, respectively. However, only 4.30 and 2.84 mg/g Pb were adsorbed via MM and BCT in stage II, proving that chemical adsorption plays a dominant role in the whole adsorption process. The adsorption process eventually enters a dynamic equilibrium adsorption stage (stage III), where the adsorption and desorption process of Pb ions reach equilibrium, and the residual Pb concentration does not diminish to zero. The Pb equilibrium adsorption capacity ($q_e$) of MM and BCT in the whole adsorption process is quantitatively determined as 14.220 and 32.934 mg/g, respectively.

As shown Fig. 2f, g, the fitting coefficient $R^2$ of the pseudo-second-order kinetic model of BCT and MM are both greater than 0.99. Furthermore, the calculated $q_e$ of BCT and MM, which are 32.48 and 14.13 mg/g, respectively, is close to the experimental results of 32.934 and 14.220 mg/g. Therefore, the pseudo-second-order kinetic model is suitable for describing the adsorption process predominated by chemisorption[32]. Based on the fitting results, the pseudo-second-order adsorption rate constants $K_{2\text{-}BCT}$ and $K_{2\text{-}MM}$ are further deduced to be 0.0690 and 0.0306 g·(mg/min), respectively. Under the same temperature conditions, $K_{2\text{-}BCT}$ is nearly twice as large as $K_{2\text{-}MM}$, which means that BCT exhibits a more excellent Pb ions adsorption rate and a lower adsorption barrier on anchoring sites. This is due to the fact that the grafted -CONH-R-$NH_2$ on BCT has multi-Pb anchoring sites, including -CONH, -$NH_2$ and -CONH-R-$NH_2$, surpassing the MM's surface with only -COOH sites.

## Pb leakage assessment of solar panels under real-world scenarios

The significant risks associated with Pb leakage from PSCs are due to two main routes: (i) Breakage of operating PSCs resulting in Pb leakage; (ii) The end-of-life perovskite panels as electronic hazardous wastes causing lasting contamination to the ecosystem. As shown in Fig. 3a, b, BCT was integrated onto solar panels with the structure of indium tin oxide (ITO)/tin oxide ($SnO_2$)/FA-based perovskite/Spiro-OMeTAD/

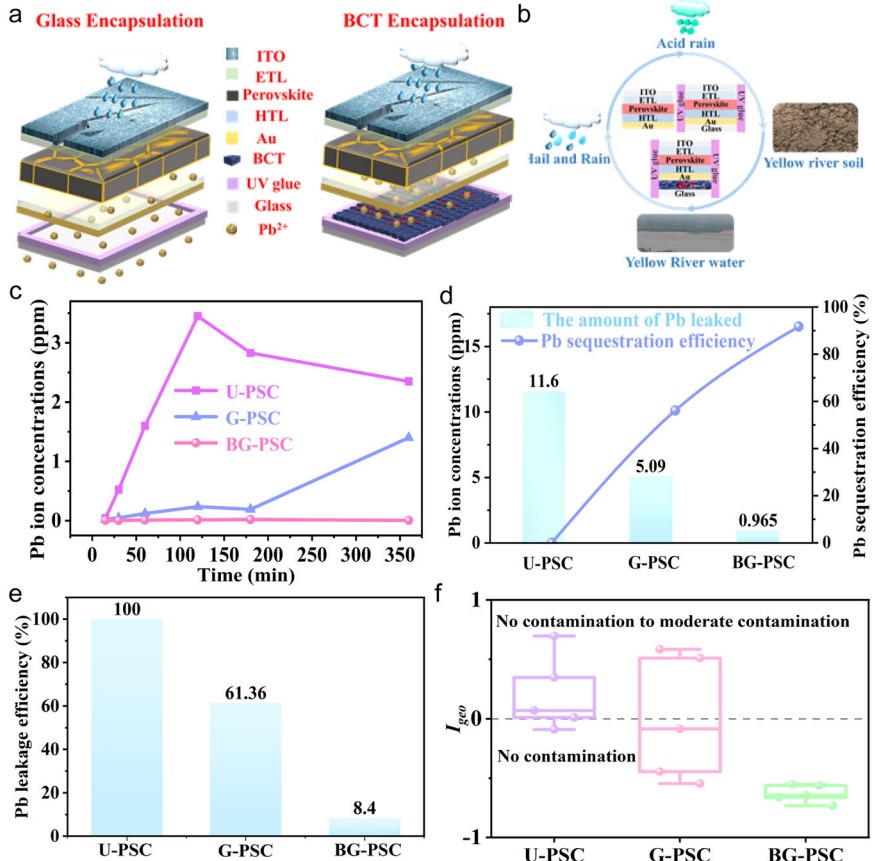

**Fig. 3 | Pb leakage assessment of damaged PSCs under real-world scenarios.**
**a** Schematic of encapsulating with glass and BCT to prevent Pb leakage when the PSCs is subjected to severe weather conditions (hail, heavy rain and acid rain) or the end-of-life damaged PSCs is abandoned into the ecosystem. **b** The Pb leakage test of PSCs in hail, rainstorm, acid rain, Yellow River water and Yellow River soil. The encapsulation methods are embedded on the figure. **c** Water- flushing Pb leakage test results for the damaged U-PSC, G-PSC and BG-PSC. **d** Pb leakage and Pb sequestration efficiencies of the damaged U-PSC, G-PSC and BG-PSC under the extreme weather simulation (acid rain (pH = 4.2) after hail impact). **e** Pb sequestration efficiencies of the damaged U-PSC, G-PSC and BG-PSC in the Yellow River water. **f** The evaluation results of end-of-life damaged U-PSC, G-PSC and BG-PSC to Yellow River soil based on $I_{geo}$.

gold (Au) to assess its capability in suppressing Pb leakage from operating solar panels when subjected to worst-case attacks such as hail and acid rain.

Hail impact simulation of PSCs were carried out according to ASTM E1038 photovoltaic panel test standard. As shown in Supplementary Fig. 12, both devices covered with pure glass (G-PSC) and a BCT layer (BG-PSC) exhibited star-shaped cracks under external collisions. The damaged panels were then submerged in flowing DI water to simulate heavy rainfall flushing right after hail attack. As shown Fig. 3c, the unencapsulated devices (U-PSC) show quick Pb leakage, with Pb ion concentration in DI water increasing to about 3.45 ppm in the first 120 min. After that, the Pb ion concentration in the aqueous solution showed a slight descending trend, which might correspond to the sedimentation of microaggregates induced by the complexation process between $PbI_2$ and $DMSO^{33}$. For G-PSC, the leakage Pb concentration reached 1.4 ppm after 6 h of soaking, and more than 40% Pb still leaked into DI water, suggesting that water permeating through the cracks indeed resulted in undesired Pb contamination. In notable contrast to the U-PSC, only 4.8 ppb Pb ions leaked from BG-PSC, achieving a Pb sequestration efficiency of 99.86%, which is below the drinking water safety level according to the WHO Guidelines for Drinking Water Quality. This indicates that hail shocks would not cause significant Pb leaking issues for the BG-PSC.

When PSCs are subjected to external impacts such as hail during practical application, the leaked Pb components also tend to flow to the backside along the cracks within the damaged PSCs by gravity

due to the modules being installed at a large inclination angle to optimize sunlight exposure for that location (Supplementary Fig. 13). Besides, we conducted additional experiments to compare the effect of different installation positions of functional materials on Pb leakage suppression performance under extreme destructive conditions. First, the mature acrylate ultraviolet curing resin was selected to package on light- accepting sides of PSCs, which have demonstrated with excellent capability of Pb leakage suppression in the previous works, and the BCT was installed on the backside of PSCs (BG-PSC) for comparison (Supplementary Fig. 14a). The U-PSC, G-PSC, resin encapsulated PSC (R-PSC) and BG-PSC were completely cracked into pieces via violent hail attacking simulation (Supplementary Fig. 14b). Then, we immersed them in DI water for 6 h and measured the concentration of leaked Pb ions. As shown in Supplementary Fig. 15, the leaked Pb ion concentrations were 7.84, 5.57 and 4.03 ppm for U-PSC, G-PSC and R-PSC, respectively. Encouragingly, only 0.876 ppm of Pb ions leaked from BG-PSC. Compared with U-PSC, G-PSC and R-PSC, which exhibit the Pb sequestration efficiency of only 0.00%, 26.79% and 48.59%, respectively, BG-PSC exhibits Pb sequestration efficiency as high as 88.83%. Therefore, it is concluded that the BCT encapsulation strategy displays much higher Pb sequestration efficiency compared with the conventional encapsulation strategy of packing the material on light-accepting glass side, even when the devices are cracked into pieces (Supplementary Fig. 16). This is mainly attributed to the advantage of large adsorption capacity and fast adsorption rate for BCT materials.

**Table 1 | The pollution level evaluation of three types of devices to Yellow River soils corresponding to $I_{geo}$, $P_i$ and $P_N$**

| Sample type | Average $I_{geo}$ value | Pollution level | Average $P_i$ value | Pollution level | Evaluation $P_N$ value | Pollution level |
|---|---|---|---|---|---|---|
| U-PSC | 0.207006 | No contamination to moderate contamination | 1.767044 | Potential contamination | 2.1256 | Moderate contamination to severe contamination |
| G-PSC | 0.004346 | No contamination to moderate contamination | 1.586362 | Potential contamination | 1.9467 | No contamination to moderate contamination |
| BG-PSC | −0.62977 | No contamination | 0.970456 | No contamination | 0.9969 | Still clean contamination |

Considering that PSCs might be practically applied in areas with serious acid rain stimulus, the nitric acid solution with a pH = 4.2 was titrated on the damaged panels at a flow rate of 10 mL/h for 1 h to simulate acid rain erosion. The contaminated water passed through the damaged devices was collected in a centrifuge tube. (The testing equipment shown in Supplementary Fig. 17). As shown in Fig. 3d, 11.6 and 5.09 ppm of the Pb ions were detected in contaminated water for U-PSC and G-PSC, respectively. Here, the Pb sequestration efficiency of U-PSC is referred to as 0%. However, only 0.965 ppm of Pb leaks from BG-PSC, achieving a Pb sequestration efficiency of 91.68%, which is due to the excellent mesoporous structure and chemical Pb chelating behavior of BCT. Moreover, we performed experiments to verify the Pb immobilization efficiency of cracked U-PSC, G-PSC and BG-PSC in an aqueous condition with pH of 10 and 11, respectively (Supplementary Fig. 18). As shown in Supplementary Fig. 18a, the leaked Pb ion concentrations for U-PSC and G-PSC are 7.82 and 2.58 ppm, respectively, at pH = 10 aqueous condition. In contrast, only 0.523 ppm of Pb ions leaked from BG-PSC, achieving 93.31% of the Pb sequestration efficiency. Meanwhile, in pH of 11 aqueous condition, compared with the Pb leakage concentration of 6.27 ppm for U-PSC, the Pb leakage concentration of BG-PSC was only 0.461 ppm, achieving a Pb sequestration efficiency of 92.64%, whereas G-PSCs had Pb leakage concentration and Pb leakage rate as high as 2.17 ppm and 34.61%, respectively (Supplementary Fig. 18b). It is noted that the BCT package can still exhibit over 90% Pb immobilization efficiency even in high pH aqueous conditions. To conduct systematic and reliable Pb leakage assessment, the Pb leakage level of damaged end-of-life PSCs in Yellow River water and Yellow River soil was measured. The samples of Yellow River water and Yellow River soil were collected from Zhengzhou, China (Supplementary Fig. 19).

To explore whether BCT can maintain high-efficiency Pb chelation behavior in complex components water, the leakage Pb concentration test on devices damaged by hail in the Yellow River water was conducted via ICP-OES. The Pb leakage of U-PSC in the Yellow River water was normalized to intuitively observe the differences in the Pb sequestration behavior between the three device structures. The Pb leakage rate of U-PSC was defined as 100%. As shown in Fig. 3e, compared with U-PSC (100%), the Pb leakage rate of BG-PSC was only 8.4%, whereas G-PSCs had a leakage rate as high as 61.36%. However, as shown in Supplementary Fig. 20 and Supplementary Table 2, compared with U-PSC (100%), the Pb leakage rates of G-PSC and BG-PSC in the DI water are only 40.56% and 0.14%, respectively. The Pb leakage rates of G-PSC and BG-PSC in the Yellow River water are 61.38% and 8.40%, respectively. It is worth noting that the Pb leakage rates of BG-PSC and G-PSC in Yellow River water increase by 8.26% and 20.82%, respectively. It can be seen that organic components, heavy metals, microbial populations and silt in the Yellow River water affect the adsorption capacity of BCT, thus resulting in an increase in Pb leakage of broken end-of-life devices. Artificially simplified Pb leakage sequestration tests in a lab scenario cannot accurately simulate the Pb leakage situation of PSCs in the real-world environments. Proposing additional procedures to measure Pb diffusion levels in real-world samples is more objective and reliable for evaluating the potential ecological pollution risk.

Once end-of-life Pb-containing perovskite photovoltaic modules are abandoned in the ecological environment, abundant toxic heavy metal Pb compounds will also be precipitated on the soil and cause persistent Pb pollution. Therefore, the surrounding land soils from the Yellow River were further collected as test samples (Supplementary Fig. 21). The characteristic evaluation parameters, including the index geo-accumulation method ($I_{geo}$), single-factor pollution index technique ($P_i$), and Nemerow integrated pollution index ($P_N$) were proposed using standard heavy metal pollution assessment methods to comprehensively analyze the Pb contamination levels in soils (See Supplementary Note 3 for detailed calculation Equation.)[34–36]. The classification criteria of heavy metal pollution assessment in soils and the actual assessment results of three types devices corresponding to $I_{geo}$, $P_i$ and $P_N$ values are summarized in Supplementary Tables 3–5 and Table 1, respectively. The average $I_{geo}$ values of U-PSC and G-PSC are 0.207006, and 0.004346, respectively, both of which can be graded into no contamination to moderate contamination levels, as shown in Fig. 3f. Encouragingly, the values of BG-PSC ranged between −0.73151 and −0.55254, which were assessed as no contamination levels, confirming Pb pollution risks can be effectively eliminated once the Pb-based end-of-life devices are discarded into the ecological environment. Besides, the alleviation of Pb pollution for the Yellow River soil are also further confirmed based on the assessment results of $P_N$ and $P_i$, as shown in Supplementary Figs. 22 and 23. Compared with U-PSC and G-PSC, BG-PSC exhibits extraordinary eco-friendly properties towards Yellow River soil.

## The compatibility validation of BCT with the photovoltaic performance and long-term device stability

Subsequent tests for validating the material compatibility with the photovoltaic performance and long-term device stability are essential for real-world application. The current density-voltage ($J$–$V$) curves and corresponding external quantum efficiencies (EQEs) of U-PSC and BG-PSC were measured individually to evaluate the influence of packaging on photovoltaic performance of PSCs, as shown in Fig. 4a, b. The $J$–$V$ characteristics of U-PSC and BG-PSC measured under reverse or forward scanning direction exhibit negligible hysteresis (Supplementary Fig. 24). The PCE statistical data based on 17 perovskite solar panels before and after encapsulation are plotted in Fig. 4c. The overlap distribution of device efficiencies with small variation and the overlap of $J$-$V$ and EQE curves indicated that the encapsulated BCT did not exert a negative impact on the photovoltaic performance. In order to evaluate the stability of PSCs under international summit on organic photovoltaic stability (ISOS) standards, the stabilities of U-PSC, G-PSC and BG-PSC were performed under continuous light at AM1.5 G, 100 mW/cm² with 60 ± 5 °C, 50% relative humidity at 25 °C and 85% relative humidity at 25 °C, respectively (Supplementary Fig. 25). As shown in Fig. 4d, it was noted that the BG-PSC was able to maintain its initial PCE of 81.49% under AM1.5 G, 100 mW/cm² at 60 ± 5°C over 360 h, whereas the PCE of U-PSCs and G-PSCs dropped dramatically to 43.65% and 36.62%, respectively. This indicated that dark BCT materials with light adsorption property may be able to partially shield against damage from thermal radiation originating from the secondary

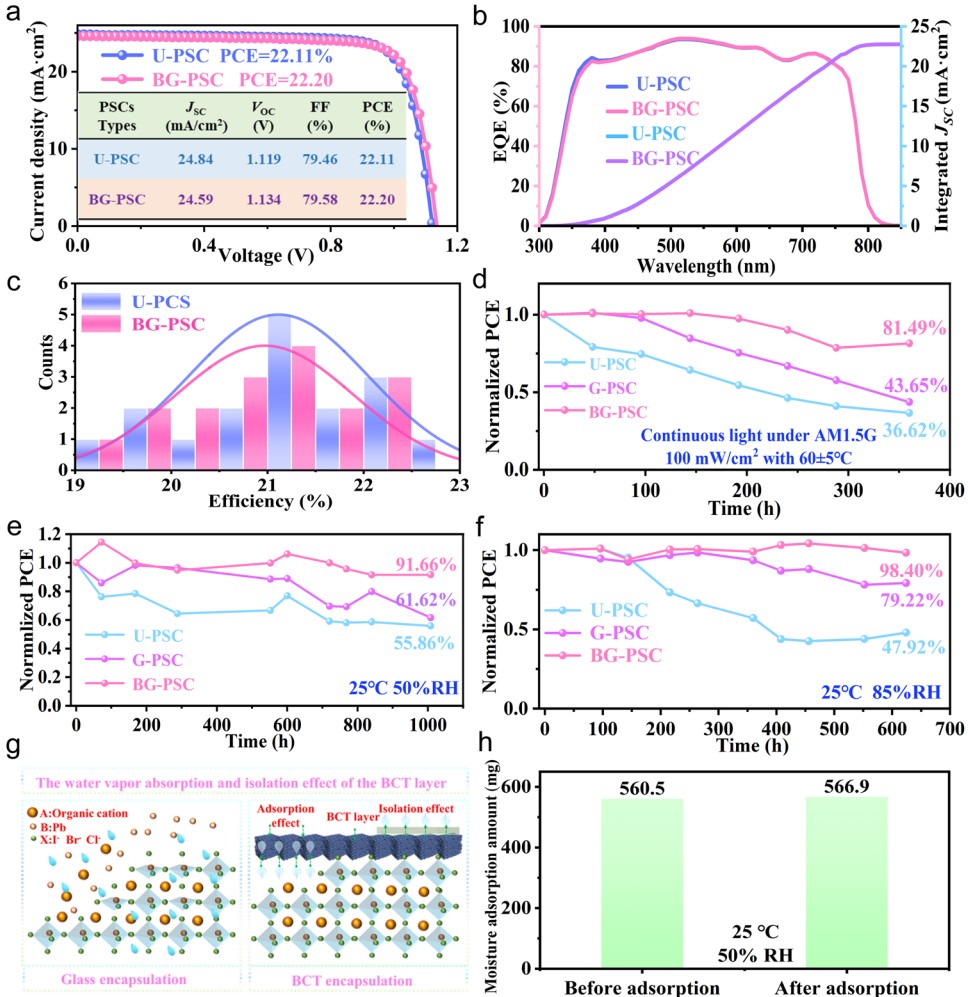

**Fig. 4 | The compatibility validation of BCT with the photovoltaic performance and long-term device stability. a** The *J-V* curves of U-PSC and BG-PSC in reverse scans under AM 1.5 G illumination. **b** The EQE spectra of U-PSC and BG-PSC, respectively. **c** Histogram of the number of cells as a function of PCE of U-PSC and BG-PSC.17 devices were used for statistics. **d** Normalized PCE of U-PSC, G-PSC and BG-PSC under continuous light at AM1.5 G, 100 mW/cm² with 60 ± 5 °C.

**e** Normalized PCE of U-PSC, G-PSC and BG-PSC stored in a moisture environment with 50% RH for over 1000 h at 25 °C. **f** Normalized PCE of U-PSC, G-PSC and BG-PSC stored in a moisture environment with 85% RH for over 600 h at 25 °C. **g** Schematic of the water vapor adsorption behavior of BCT. **h** Moisture adsorption capacity of BCT layer.

reflection of light from the side of Au electrode during continuous light exposure. When the U-PSC, G-PSC and BG-PSC were exposed to a moisture environment with 50% RH for over 1000 h at 25 °C, the PCE for U-PSC and G-PSC decreased dramatically to 56% and 62% of the initial PCE after the 1000-h test respectively, which can be attributed to the degradation of the perovskite layer due to the moisture entering into the device through the ultraviolet (UV) curing adhesive gap. The PCE of BG-PSC could retain 92% of the initial PCE (Fig. 4e). As shown in Fig. 4f, the PCEs of the U-PSC and the G-PSC dropped to 47.92% and 79.22% of their initial PCEs at a moisture environment with 85% RH for over 600 h at 25 °C, respectively. It is worth noting that the BG-PSCs exhibit promising stability and maintained 98.40% of their initial PCE, due to the synergistic role of the BCT in terms of internal secondary water vapor adsorption and external water vapor isolation, preventing the erosion impact of moisture on perovskite material (Fig. 4g and Supplementary Fig. 26). To verify the water vapor adsorption behavior, the BCT was placed at an ambient environment with 50% RH for 24 h. It was discovered that the weight increased from 560.5 mg to 566.9 mg, proving excellent water vapor absorption characteristics of BCT (Fig. 4h). These results proved that the BCT has excellent compatibility with

photovoltaic performance and can enhance the long-term stability of the PSCs.

## Pb recycling engineering under sustainable closed-loop management

Further study of this work focused on creating a close-loop Pb management procedure, which needs a top priority to ensure low materials cost and avoid secondary pollution from the Pb adsorption materials. The cyclic process was rationally divided into four critical steps, including I Pb precipitation, II Pb adsorption, III Pb desorption and IV Pb recycle, as shown in Fig. 5a. In procedures I-II, the BCT can effectively adsorb Pb ions from the solution with an adsorption rate of 99.77% via chemical coordination bonding between Pb ions and RCO-NH-R, $RNH_2$, $CONH$-R-$NH_2$, as shown in Supplementary Fig. 27a. In procedure III, the Pb release process was mainly dominated by secondary protonation reaction. In a highly $HNO_3$ solution environment, the protonated hydrogen ion could be chelated by the lone pair around N element in RCO-NH-R, $RNH_2$ and $CONH$-R-$NH_2$, forming RCO-$NH_2^+$-R, $RNH_3^+$, and $CONH_2^+$-R-$NH_3$. The secondary protonation reaction destabilized the existed binding force between the RCO-NH-$CH_2$-$CH_2$-$NH_2$ and Pb ions because H processes larger electronegativity

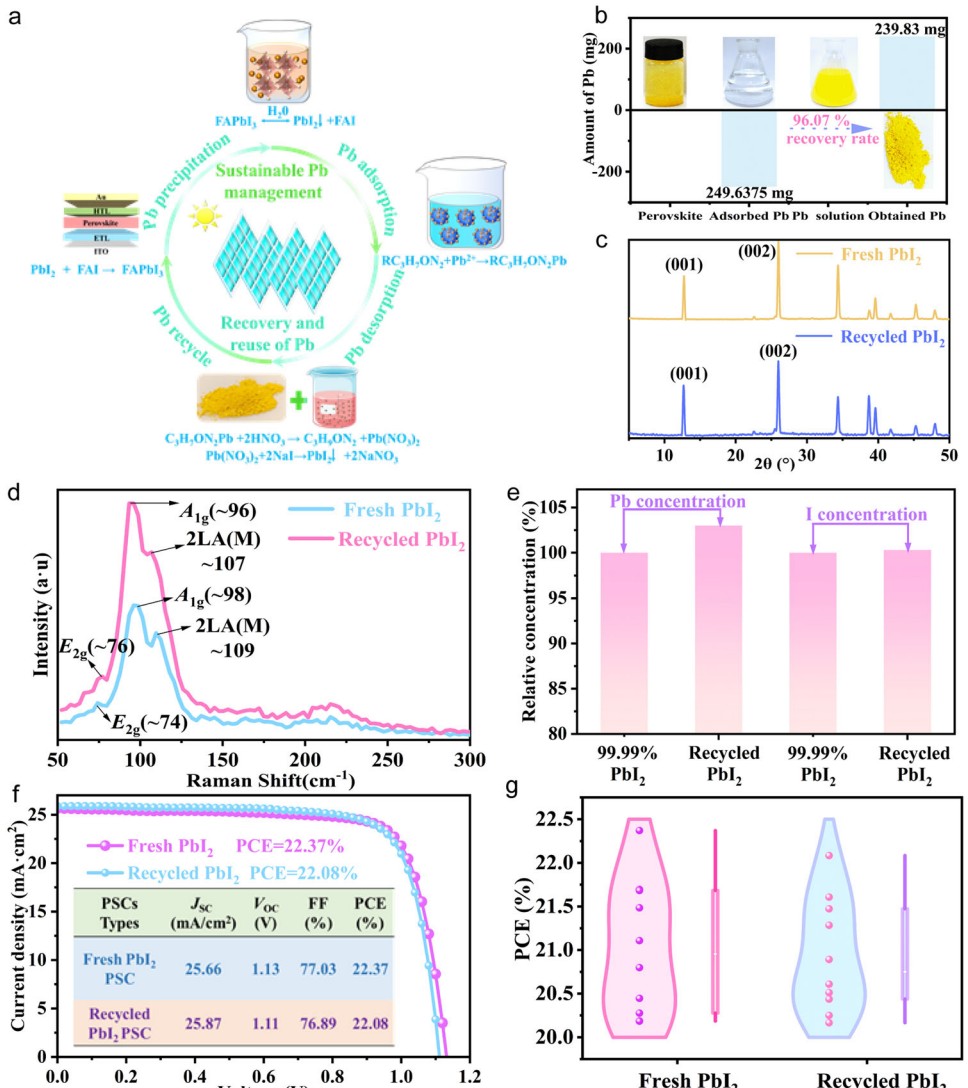

**Fig. 5 | Pb recycling engineering under sustainable closed-loop management.**
**a** Schematic of sustainable Pb management of perovskite solar modules, including I
Pb precipitation, II Pb adsorption, III Pb desorption and IV Pb recycle. **b** Pb recycling
ratio via BCT recycle. **c** The XRD of commercial fresh PbI$_2$ and recycled PbI$_2$. **d** The
Raman spectra pattern of commercial fresh PbI$_2$ and recycled PbI$_2$. **e** Relative Pb
concentration and I concentration in DMF solution with the equivalent amount of

commercial fresh 99.99% PbI$_2$ and recycled PbI$_2$, where the commercial fresh
99.99% PbI$_2$ was used as the reference. Pb and I concentration were determined via
ICP-OES. **f** The *J-V* curves of PSCs based on commercial fresh 99.99% PbI$_2$ and
recycled PbI$_2$ in reverse scans under AM 1.5 G illumination. **g** Violin Plot of the PCE
of PSCs which prepared by the commercial fresh 99.99% PbI$_2$ and recycled PbI$_2$. 10
devices were used for statistics.

than Pb (H (2.2) > Pb (1.9))[37], resulting in releasing the captured Pb ions
as shown in Eqs. (4)–(6) and Supplementary Fig. 27b, c.

$$R_2CONHPb^{2+} + 2HNO_3 \rightarrow RCO - NH_2^+ - R + Pb(NO_3)_2 + H^+ \quad (4)$$

$$RNH_2Pb^{2+} + 2HNO_3 \rightarrow RNH_3^+ - R + Pb(NO_3)_2 + H^+ \quad (5)$$

$$(RC_3H_7ON_2)_2Pb + + 2HNO_3 \rightarrow 2RCO - NH_2^+ - CH_2 - CH_2 - NH_3^+ + Pb(NO_3)_2 \quad (6)$$

We have demonstrated the protonation process via characterizing the FTIR of the mixed solution of CONH-R-NH$_2$ with PbI$_2$ (referred
as CONH-R-NH$_2$@PbI$_2$) and the mixed solution of CONH-R-NH$_2$, PbI$_2$
with HNO$_3$ (referred as CONH-R-NH$_2$@PbI$_2$ + HNO$_3$), respectively. As
shown in Supplementary Fig. 28, CONH-R-NH$_2$@PbI$_2$ shows characteristic peaks at around 3340 cm$^{-1}$ and 3277 cm$^{-1}$, which are assigned

to the stretching vibration of -NH in -NH$_2$. And the characteristic strong
vibration absorption peak that appear at 1550 cm$^{-1}$ is related to the
stretching vibration of the -C=O in the -CONH group. In comparison
with CONH-R-NH$_2$@PbI$_2$, it is found the peak positions of the
stretching vibration of -NH in CONH-R-NH$_2$@PbI$_2$ + HNO$_3$ samples
obviously shift from 3340 cm$^{-1}$ and 3277 cm$^{-1}$ to 3228 cm$^{-1}$and
3165 cm$^{-1}$ in FIIR spectra, respectively. The new peaks regarding the
stretching vibration of protonated amine groups (-NH$_3^+$) at around
2823 cm$^{-1}$ and 2596 cm$^{-1}$ can be clearly observed in CONH-R-
NH$_2$@PbI$_2$ + HNO$_3$, indicating that the -NH$_2$ group was protonated
via the H$^+$ ionized by HNO$_3$[38]. Meanwhile, the new peak appears at
around 2535 cm$^{-1}$ and 1514 cm$^{-1}$, which are assigned to the stretching
vibration of protonated amide(-NH$_2^+$) and protonated amide II, manifesting the -CONH was protonated by H$^+$[39,40]. Besides, the peak positions of the stretching vibration of -C=O also obviously shifts from
1550 cm$^{-1}$ to 1606 cm$^{-1}$. The reason is attributed to that H$^+$ ionized by
HNO$_3$ coordinates with the lone pair on the N atom on the -CONH, thus
breaking the conjugated system of the -CONH- and making the -C=O

bond shift to high frequency due to its enhanced double bond properties[41,42]. The above results show that the Pb desorption is achieved based on the protonation of CONH-R-NH$_2$ molecular configuration by HNO$_3$.

Considering whether the PbI$_2$ can be cyclically utilized, sustainable chemical exchange reactions are designed for desorbing Pb in step III, achieving 96.07% of recovery efficiency of yellow PbI$_2$ powder (Supplementary Fig. 29, Fig. 5b and Supplementary Movie 1). In order to explore whether the recycled PbI$_2$ purity meets the commercial application standard, the crystal structure and ingredient composition of recycled PbI$_2$ were further characterized by X-ray diffraction (XRD), Raman Spectrometer, and FTIR, respectively. As shown in Fig. 5c, the recycled PbI$_2$ exhibits matched XRD patterns with commercial fresh PbI$_2$. The most prominent diffraction peaks of (001) and (002) crystal planes are both observed for the commercial fresh PbI$_2$ and recycled PbI$_2$[43]. Meanwhile, the recycled PbI$_2$ shows the coincident molecular framework as commercial PbI$_2$, which can be verified via Raman characterization. The strong characteristic Raman bands are observed at 74 (E$_{2g}$ vibration peak, that is assigned as the 4H phase of our PbI$_2$ flakes)[44], 98, 109 cm$^{-1}$ (A$_{1g}$ and 2LA(M) correspond to overtone of the LA phonon at the M point of first Brillouin zone, respectively)[45,46] in commercial fresh PbI$_2$ and 76, 96, 107 cm$^{-1}$ in recycled PbI$_2$ (Fig. 5d)[47]. Notably, the intensity of the Raman characteristic peak in recycled PbI$_2$ is higher as compared to the commercial fresh PbI$_2$, indicating that the recycled PbI$_2$ possess higher crystalline[48]. The slight shift of the peaks is mainly subjected to varying lattice stress caused by size and shape of powders[49]. Meanwhile, there are no additional peaks of impurity chemical groups appear in FTIR spectra (Supplementary Fig. 30). The purity of recycled PbI$_2$ and commercial fresh PbI$_2$ (99.99%) were further quantified ICP-OES via analyzing Pb and iodine(I) contents that were dissolved in DMF solution.

As shown in Fig. 5e, there exist a small deviation from stoichiometric ratio of Pb and I element in recycled PbI$_2$ sample. The recycled PbI$_2$ has 103% of relative Pb concentration and 100.3% relative I concentration compared with commercial PbI$_2$ (set as a standard control sample with 100% concentration). The actual chemical ratio of I to Pb is 1.93449, indicating the deficiency of the I component. Therefore, the recycled samples should be rectified to PbI$_{1.93449}$. The excess stoichiometric of Pb may decrease the PCE performance of the devices that fabricated based on recycled PbI$_2$ due to production of more defect states. The final step IV is realized via refabricating the PSCs based on recycled PbI$_2$. It shows that FA-based PSCs prepared by recycled PbI$_2$ display 22.08% PCE with an open circuit voltage ($V_{OC}$) of 1.11 V, a short-circuit current density ($J_{SC}$) of 25.87 mA/cm$^2$, a fill factor (FF) of 76.89%, which is almost close to the device fabricated on commercial fresh PbI$_2$ achieves a PCE of 22.37% with a $V_{OC}$ of 1.13 V, a $J_{SC}$ of 25.66 mA/cm$^2$, and a FF of 77.03% (Fig. 5f). Please note that the device of $J_{SC}$ here higher than that of U-PSC and BG-PSC in Fig. 4b is attributed to the seasonal difference between March and October caused significant changes in climate temperature, humidity of experimental operating environment and glove box temperature during the preparation of PSCs by two-step method. The crystalline quality of perovskite thin films, such as grain size and density of defect states, is affected, thus influencing device performance[50,51]. The statistical PCE based on 10 perovskite solar devices prepared with commercial fresh PbI$_2$ and recycled PbI$_2$ are plotted in Fig. 5g. The average PCE values of the two types of devices are 21.02% and 20.93% respectively. The standard difference factor (SDF) was further calculated, which is defined by SDF(%) = $\left(1 - \frac{\text{PCE of PSCs based on recycled PbI}_2}{\text{PCE of PSCs based on commercial PbI}_2}\right)$%. Consequently, the SDF for devices manufactured with recycled PbI$_2$ is only 0.42%, verifying the recycled PbI$_2$ with high purity almost dose not compromise PSCs' photovoltaic performance.

Overall, BCT materials offer an efficient way to reduce Pb contamination during the actual operating process of PSCs while reducing the cost of PSC production, as shown in Supplementary

Fig. 31. They also offer a more environmental friendly alternative to polymer materials and will promote the green commercialization of PSCs.

## Discussion

In summary, inspired by the ensnaring prey behavior of spiders, we strategically design BCT to address current challenges regarding Pb contamination and the operational stability of PSCs. The synergistic Pb capture mechanisms based on physical adsorption and chemical chelation are unraveled. The microcosmic capillary adsorption process in the reaction chambers of MM is revealed. Besides, the synthetic reaction of dehydration condensation for implanting functional molecular (CONH-R-NH$_2$) and the impact of molecular configuration on Pb capturing capacity are in-depth analyzed. Real-world Pb contamination assessment is also conducted, which shows that the BCT exhibits impressive Pb shielding capacity. The leaked Pb ion concentrations from the end-of-life PSCs are accurately quantified in Yellow River and surrounding soils for real-world Pb contamination assessment. Encouragingly, the BCT exhibits impressive Pb shielding capacity, where the extent of Pb leakage is limited as low as safe contamination levels even in the worst-case scenarios such as hail and acid rain attacking. More importantly, devices with the BCT encapsulation show high operation stability with less than 8% relative efficiency loss under 25 °C and 50% RH for over 1000 h storage. Moreover, a rational Pb recycling approach based on chemical ion exchange reaction is developed, which provides the feasibility for closed-loop management from Pb precipitation to Pb recycling. The purity of recycled PbI$_2$ is comparable to commercial 99.99% PbI$_2$, and the refabricated device shows a PCE of 22.08% without significant photovoltaic performance drop compared with fresh counterparts. This work provides deep insights into the microscopic structure-property-performance relationship in multifunctional Pb adsorption materials and paves the way for accelerating PSCs to commercial applications.

## Methods

### Materials

Mesoporous matrixes (MM) were purchased from Henan MeiYuan Water Purification Material Comp. Ltd. Ethylenediamine (EDA, 99.0%) sulfuric acid (H$_2$SO$_4$, 18.4 mol/L) and nitric acid (HNO$_3$, 16 mol/L) were purchased from Tianjin FengChuan Chemical Reagent Technology Comp. Ltd. Indium tin oxide (ITO) (6 Ω sq$^{-1}$, 16.1 mm × 16.1 mm) was purchased from South China Science & Technology Comp. Ltd. Lead iodide (PbI$_2$, 99.99%), formamidinium iodide (FAI, 99.99%), methylammonium bromide (MABr, 99.5%), methylamine hydrochloride (MACl, 99.5%), 2,2',7,7'-tetrakis(N,N-di-p-methoxyphenyl-amine)−9,9'-spirobifluorene(spiro-OMeTAD, 99.8%), bis(trifluoromethane) sulfonamide lithium salt (Li-TFSI, 99%), and 4-tert-butylpyridine (TBP, 96%) were purchased from Xi'an Polymer Light Technology Corp. SnO$_2$ solution (15 wt% in H$_2$O colloidal dispersion) was purchased from Alfa Aesar. N, N-dimethylformamide (DMF, 99.8%), dimethyl sulfoxide (DMSO, 99.7%) and chlorobenzene (CB, 99.8%) were purchased from Sigma-Aldrich. Au was purchased from ZhongNuo Advanced Material (Beijing) Technology Co., Ltd. Lead ion detection test paper was purchased from Zhejiang LuHeng Environment Technology Comp. Ltd. Ultraviolet curing adhesive (UV glue, 170 cps) was purchased from Beijing New Material Technology Comp. Ltd. Low temperature conductive carbon paste (40 Ω) was purchased from Shenzhen Deliou Science and Technology Development Comp. Ltd. Isopropanol (IPA, 99.5%) and sodiumiodide (NaI,99.5%) were purchased from Aladdin. All materials above were used as received. The samples of Yellow River water and Yellow River soil were collected from Zhengzhou, China.

### Chemical synthesis of the biomimetic cage traps (BCT)

Firstly, in initial state, MM was ground into fine powder. Then, in ultrasound-assisted structural separation stage, 300 mg of purchased

MM was transferred to a 100 mL beaker with 50 mL flowing deionized water (DI water) and reacted at 25 °C for 90 min. Finally, in surface chemical reconfiguration stage, Different volumes of EDA was drop into the beaker and reacted at 40 °C for 150 min to obtain BCT mixture. The BCT mixture was separated via vacuum filtration using Buchner funnel filter paper with porosity of 0.45 mm and thoroughly washed with a 1:1 mixture of ethanol and water. Then, the BCT solid was obtained via drying in a vacuum oven at 25 °C for 12 h.

## Device fabrication

ITO glasses substrates (6 Ω sq$^{-1}$, 16.1 mm × 16.1 mm) were sequentially cleaned via sequential ultrasonication in detergent, DI water, acetone, ethanol for 15 min respectively. The ITO glasses substrates were dried with dry nitrogen ($N_2$) and treated by oxygen plasma for 15 min. The $SnO_2$ aqueous colloidal dispersion was mixed with DI water in the volume ratio of 1:6 and diluted by ultrasonication at room temperature for 15 min. Then, the $SnO_2$ was deposited onto glasses substrates to generate compact $SnO_2$ as the electron transport layer via spin-coating with 4000 rpm for 30 s and annealed in ambient air under 180 °C for 20 min.

The perovskite film was fabricated based on the two-step deposition method. Firstly, 1.3 M of $PbI_2$ in DMF/DMSO (9.5:0.5) solvent (50 μL) was spin-coated onto $SnO_2$ at 1500 rpm for 30 s and annealed at 65 °C for 60 s, then cooled to room temperature. Secondly, 150 μL mixture solution of FAI/MABr/MACl (60:6:6 mg in 1 mL IPA) was spin-coated onto the $PbI_2$ film with spin rate of 1500 rpm for 30 s. The perovskite precursor film was transferred into ambient air (35–40% relative humidity (RH)) for thermal annealing at 140 °C for 10 min to obtain high-quality perovskite as the active layer.

Then, spiro-OMeTAD as hole transporting layer was coated onto the perovskite film with a spin rate of 4500 rpm for 30 s, which consisted of 72.3 mg mL$^{-1}$ spiro-OMeTAD, 28.8 μL TBP, 17.5 μL Li-TFSI (520 mg mL$^{-1}$ Li-TFSI in ACN) and 1 ml CB. Finally, 70 nm Au or (10 nm $MoO_3$ + 60 nm Au) film was thermal evaporated as electrode via a shadow mask (0.04 cm$^2$)

## Device encapsulation

BCT was mixed with low-temperature conductive carbon paste at a ratio of 2:1, and was coated on the glass substrate. The substrate with BCT coating layer was placed on the hot-plate and annealed at 150 °C for 30 min to make BCT hot-press on the glass substrate. After cooling to room temperature, the substrate with BCT coating layer was transferred into the $N_2$ glove box. Then, the pure glass substrate and substrate with BCT coating layer were encapsulated onto the perovskite solar cells (PSCs) via coating UV glue around the device.

## Characterization

Fourier transform infrared (FTIR) spectra dates were measured by Bruker's INVENIO R spectrometer. The X-ray photoelectron spectroscopy (XPS) data were carried out using Shimadzu AXIS Supra. The surface morphologies and energy dispersive spectroscopy (EDS) of the samples were characterized by the scanning electron microscopy (Thermo Scientific/Helios G4 CX). The roughness of MM and BCT were measured by 3D optical profilometer (Sensofar). The external surface area of MM and BCT is evaluated via Brunauer–Emmett–Teller surface area and pore size analyzer. Ultraviolet-visible (UV-vis) absorption spectra was measured by a UV-Vis-NIR spectrometer (Cary5000 UV-Vis-NIR). The Pb ion concentrations were measured via inductively coupled plasma emission spectrometer (ICP-OES) and graphite furnace atomic absorption spectrometry (GFAAS), respectively. The current density-voltage (J-V) characteristics of the perovskite solar cells (PSCs) were obtained under AM 1.5 G, 1 sun simulated irradiation at 100 mW/cm$^2$ (Enlitech, SS-X180R). The external quantum efficiency (EQE) spectrum was obtained via solar cell spectral response measurement system (Enlitech, QE-R). X-ray diffraction (XRD) patterns were conducted by a multifunctional X-ray diffractometer (Smartlab) with Cu Kα radiation ($\lambda$ = 1.54184 Å). Raman spectra was measured on the LabRAM HR Evolution Raman spectrometer via using the 532 nm wavelength of an argon ion laser as the excitation source.

## Pb adsorption and leakage test

For the adsorption dynamics tests of Pb$^{2+}$ by the MM and BCT in aqueous solution, 0.3 g MM and BCT was poured into 60 mL of 167 ppm Pb perovskite powder solution without stirring. A series of samples were extracted from the lead solution at different time intervals ranging from 1 to 180 min at room temperature. The Pb ion concentration in samples were measured by ICP-OES instrument to determine.

In the Pb leakage test, all the dimensions of the three types devices are 16.1 mm × 16.1 mm. For the test of heavy rainfall flushing, damaged unpackaged perovskite solar cell (U-PSC), glass encapsulated perovskite solar cell (G-PSC) and BCT encapsulated perovskite solar cell (BG-PSC) were placed in a 100 mL beaker, and 60 mL of DI water was added to the beaker to achieve complete immersion of the damaged device. Then contaminated 5 mL DI water in the beaker at 15 mins, 30 mins, 1 h, 2 h, 3 h, and 6 h were taken, respectively. The Pb ion concentration in the contaminated DI water was quantitatively measured by ICP-OES. For the Pb leakage test of completely broken PSCs, the completely broken U-PSC, G-PSC, resin encapsulated PSC (R-PSC) and BG-PSC were placed in a 50 mL beaker, and 10 mL of DI water was added to the beaker to achieve complete immersion of the completely broken device. Then contaminated 5 mL DI water in the beaker at 6 h was taken. The Pb ion concentration in the contaminated DI water was quantitatively measured by ICP-OES. In the Pb leakage test of PSCs under acid rain condition, U-PSC, G-PSC and BG-PSC were placed in the funnel at a 45° inclination relative to the ground, as shown in Supplementary Fig. 17. Then, a nitric acid solution with a pH value of about 4.2 was prepared and titrated on the device at a flow rate of 10 mL/h for 1 h to simulate extremely heavy acid rainfall. The contaminated Pb-containing solutions that passed through the funnel were collected in glass vials for ICP-OES testing. In the Pb leakage test of PSCs at high pH aqueous condition, damaged U-PSC, G-PSC and BG-PSC were placed at aqueous condition with pH = 10 and pH = 11, respectively. The contaminated 5 mL alkaline water at 6 h was taken. The Pb ion concentration in the contaminated DI water was quantitatively measured by ICP-OES.

In the real-world Pb contamination assessment of end-of-life PSCs, the samples of Yellow River water and Yellow River soil with Pb background values of 0.0101 ppm and 17.6 mg/kg were taken from Zhengzhou, China. For the Pb leak test of the damage end-of-life device in the Yellow River water, the damaged U-PSC, G-PSC and BG-PSC were placed in a 100 mL beaker containing 30 mL of Yellow River water to achieve complete immersion of the device, respectively. After 6 h, a series of samples of the contaminated Yellow River water in the beaker was taken and tested for the Pb ion concentration by ICP-OES. In the Pb contamination assessment of damaged end-of-life PSC to the Yellow River soil, the Yellow River soil was mixed evenly and divided into three pots to simulate the soil environment. U-PSC, G-PSC and BG-PSC damaged by the hail crushing simulation were buried in the soil, as shown in the Supplementary Fig. 21. Then DI water was poured daily at the location where the devices were buried. The soils below the buried devices at 5 cm were sampled on days 1, 7, 14, 21, and 30 based on the diagonal sampling method. And the total Pb in the soil was determined by GFAAS. Then, the characteristic evaluation parameters including index geo-accumulation method (Igeo), single-factor pollution index technique (Pi), and Nemerow integrated pollution index (PN) were proposed using standard heavy metal pollution assessment methods to comprehensively analysis the Pb contamination levels in soils.

## Stability test

The devices with 10 nm $MoO_3$ + 60 nm Au were aged under continuous light at AM1.5 G, 100 mW/cm$^2$ with $60 \pm 5$ °C, and measured J-V curve under a solar simulator (AM1.5 G, 100 mW/cm$^2$) at different time intervals to record the attenuation of efficiency. The devices with 70 nm Au were aged under 25 °C, 50% relative humidity and 25 °C, 85% relative humidity conditions in a constant temperature and humidity chamber, and measured J-V curve under a solar simulator (AM1.5 G, 100 mW/cm$^2$) at different time intervals to record the attenuation of efficiency.

## Sustainable Pb management procedure

We prepared FA-based perovskite and made it completely dissolved in 450 mL DI water by heating and stirring to prepare a perovskite powder solution. 2.5 g BCT was added into 450 mL of perovskite powder solution with 556 ppm Pb ion concentration at 100 °C and stirred for 12 h under 40 °C. Solid-liquid separation of BCT adsorbed with Pb ions and perovskite solution via filtering, obtaining BCT which adsorbed a large amount of Pb. The isolated BCT was dissolved in 200 mL of pH = 1 nitric acid ($HNO_3$) and stirred at 40 °C for 4 h to generate $Pb(NO_3)_2$. The purpose is to elute the Pb ions adsorbed on BCT. The BCT and $Pb(NO_3)_2$ were separated by filtration to obtain $Pb(NO_3)_2$ solution. Then 1 mol/L sodium iodide was added dropwise to the $Pb(NO_3)_2$ solution to form a yellow $PbI_2$ suspension, as shown in the Supplementary Fig. 29 and Supplementary Movie 1. Then, the yellow $PbI_2$ suspension was washed with IPA. The yellow $PbI_2$ powder was obtained by centrifugation and vacuum drying. Finally, the recovered $PbI_2$ was used to fabricate FA-based PSCs.

## Density function theory calculations

The ab initio calculations were performed based on the Vienna ab initio simulation package (VASP) within the framework of density functional theory (DFT) for structural optimizations and energy simulations[52]. We adopted the all-electron projector-augmented wave potential (PAW) with Perdew–Burke–Ernzerhof generalized gradient approximation as the exchange-correlation function to describe electron-ion interaction[53,54]. Meanwhile, we treated $2s^2 2p^2$, $1s^1$, $2s^2 2p^4$ configurations as valence electrons for C, H and O respectively. The energy cutoff (400 eV) for the plane-wave expansion of eigenfunctions and Monkhorst-Pack meshes of $4 \times 4 \times 1$ were selected to guarantee the better energy converges.

## Reporting summary

Further information on research design is available in the Nature Portfolio Reporting Summary linked to this article.

## Data availability

The data that support the findings of this study are available in the following repository: https://doi.org/10.6084/m9.figshare.23282813. The source data is provided with this work. Source data are provided with this paper.

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

## Acknowledgements

The authors thank the financial support from the National Key R&D Program of China (Grant No. 2018YFA0208501 (Y.S.)), the National Natural Science Foundation of China (Grant Nos. 62104216 (J.M.) and 91963212(Y.S.)) and Beijing National Laboratory for Molecular Sciences (No. BNLMS-CXXM-202005 (Y.S.)), the China Postdoctoral Innovative Talent Support Program (Grant No. BX2021271(P.L.)), Key R&D and Promotion Project of Henan Province (Grant no. 192102210032 (Y.Z.)), Opening Project of State Key Laboratory of Advanced Technology for Float Glass (Grant No. 2022KF04 (Y.Z.)), the Joint Research Project of Puyang Shengtong Juyuan New Materials Co., Ltd. (Y.Z.), and Outstanding Young Talent Research Fund of Zhengzhou University (J.M. P.L. and Y.Z.). The authors also thank the Advanced Analysis & Computation Center at Zhengzhou University for materials and device characterization support.Received: ((will be filled in by the editorial staff)) Revised: ((will be filled in by the editorial staff)) Published online: ((will be filled in by the editorial staff))

## Author contributions

Y.S., Y.Z. and J.M. supervised the project. Y.Z., J.M. and H.L. conceived the idea and designed the experiments. H.L., P.L., Y.Z. and J.M. performed BCT synthesis, device packaging, Pb leakage test and Pb recycling, most of the characterizations and curated collected data. X.L., H.Z., Y.C., Q.L., and Q.X. performed data analysis. X.L. performed first-principles density-functional theory. H.Z., Y.C. and Q.L. completed the photovoltaic performance test of the device. H.L., P.L., J.M., Y.Z. and Y.S. significantly contributed to the discussion of the data. H.L. and J.M. wrote the first draft of the paper. All authors contributed to the interpretation of data, to the discussion, to paper writing and approved the final version.

## Competing interests

The authors declare no competing interests.
