## [Peer Review File · Nature Communications]

Bioinspired cage traps for closed-loop lead management of perovskite solar cells under real-world contamination assessmentREVIEWER COMMENTS

Reviewer #1 (Remarks to the Author):

This paper proposed a closed-loop lead management system from adsorption to recycling for lead in perovskite solar cells by utilizing Pb adsorption materials called bioinspired cage traps. It also simulated the breakdown of solar cells in various real-world environments and showed the results for real-world lead leakage. These findings are of great significance as the commercialization of perovskite solar cells is becoming more likely.

In reviewing this study, there were three main areas.

- i) Development of BCT lead adsorbent material and identification of the adsorption mechanism
- ii) Simulation of the destruction situation through various real-world simulations
- iii) Long-term stability and closed-loop lead recycling

Although the three areas convey important messages in the solar cell assessment, I believe that the originality and scientific impact in each area are insufficient to satisfy the readership level of Nature Communications. Therefore, I decline publication in this journal.

The comments that follow are our rationale.

i) BCT materials

In the description of the BCT adsorbent, similarities with the mesh structure and adhesive properties of spider silk, were mentioned. However, in the case of BCT synthesized based on MM, the porous pores were simply increased from MM and the chemical reactivity with Pb was given by adsorbing EDA on them. The parent material, MM, is also a commercially available product, and the binding of EDA to Pb utilizes the very common result of Pb binding to an amine group in the organic A site. In other words, it is almost impossible to find anything new respect to materials. When I received this article first, I expected to find a semi-transparent adsorbent with a spider web-like mesh structure, but this was not the case. Neither the structural properties of the material, nor the material itself, make it appropriate to label it bioinspired.

We also discussed physical and chemical adsorption, but again, capillarity is proportional to the number of pores, and chemical adsorption does not deviate much from the already known principles. In other words, the results are predictable and do not give the reader any insight.

In the synthesis of BCT, it is not clear at what stage and by what reaction the number of pores of MM increases, so there is a lack of in-depth discussion of the specific surface area enhancement, which has a significant impact on Pb adsorption.

The results of FT-IR, XPS, etc. are predictable results that should be expected after the completion of BCT synthesis or Pb adsorption, but the manuscript is not complete enough to satisfy the answers, such as materials generation mechanism or give properties.

ii) Simulation of destruction situation of PVSK solar cells

This is the part where we can give the most originality to this research. However, there is a structural limitation in the encapsulation of BCTs by mixing them with conductive carbon. If the destruction only occurred on the light-accepting glass side, the encapsulated BCTs would not adsorb Pb enough. To solve this problem, it is thought that if double-sided application is used, there will be a sharp decrease in efficiency due to the black color of the carbon.

iii) Recycling

This part is almost impossible to give the value of originality, scientific discovery, etc. to be published in Nature Communications because its mechanism is a well established already, and demonstrated similarly so many times.

As such, this study is lacking the novelties and originalities in all three areas, and it is appropriate to publish it in a more detailed journal.

Reviewer #2 (Remarks to the Author):

See Attachment

Pb contamination risk is one the most important factors that influence the commercialization of PSCs. Luo et al. were inspired by the spider web and then implanted a multifunctional mesoporous amino-grafted-carbon net into PSCs. The study has indicated that amino-grafted-carbon net can serve as a biomimetic “cage traps” for mitigating Pb leakage. Besides, They have established a sustainable closed-loop Pb management engineering via four critical steps including Pb precipitation, Pb adsorption, Pb desorption, and Pb recycling. It's a fascinating work supported by detailed characterization and data. Even so, there are some problems that need to be clarified before accepting.

1. The author mentioned that the properties the Yellow River water affect the adsorption capacity of BCT, thus resulting in the increasing of Pb leakage of broken end-of-life devices. However, the Pb leakage rates of BG-PSC and G-PSC in the DI water only are 0.14% and 40.56%, whereas their Pb leakage rates in Yellow River water increase by 8.26% and 20.8%, respectively. Compared with that in DI water, why does the Pb leakage rates increase for BG-PSC while reduce for G-PSC in Yellow River water. Moreover, the data described in the text is inconsistent with the Supplementary Figure 13, please correct it.
2. The author proved excellent water vapor absorption characteristics of BCT (Fig. 4f) in Line 357. So, how to understand the synergistic roles in terms of internal secondary water vapor adsorption and external water vapor isolation of the BCT.
3. The purity of recycled PbI_2 is higher than the commercial fresh PbI_2 . Why does the performance of device based on recycled PbI_2 is lower than that based commercial fresh PbI_2 .
4. The intensity of XPS data in this paper needs to be normalized.

Reviewer #3 (Remarks to the Author):

This work describes Pb capturing idea for sustainable perovskite solar cells (PSCs). Bio-inspired cage traps for the lead ions were designed and its function of immobilization of lead ions was studied using acidic water. In addition, bio-inspired system was applied to PSCs, which was studied in terms of Pb immobilization ability and photovoltaic performance. This work is recommended for publication after minor revision.

1. What about Pb immobilization efficiency at high pH aqueous condition?

2. What about PV performance under 85% RH and 85 °C?

Reviewer #4 (Remarks to the Author):

This work reported a novel bioinspired sealant material, entitled "cage traps" (BCT), that takes inspiration from spider webs. This encapsulation strategy will protect the devices from the erosion of humidity and act as a blocker to adsorb lead that might leak. The working mechanism of the BCT strategy involves a synergistic approach that combines chemical chelation and physical adsorption to address lead toxicity and the instability of perovskite modules. The author also highlights the simulated real-world test conditions by assessing the lead contamination in Yellow River and surrounding soils. This research provides valuable insights into material design and prudent tests, which sets the groundwork for further research despite some controversial points. Hence, the work warrants publication in Nature Communication after addressing the following issues.

1. Since the photovoltaic performance is mentioned repeatedly in Fig. 4b and Fig. 5f, there appears to be a noticeable difference in the short-circuit current density (J_{sc}), which decreases from approximately 25.8 to 24.8 mA/cm². This discrepancy has led to some confusion regarding the possible causes of this J_{sc} loss, whether it could be attributed to the encapsulation process or if the performance analysis of the devices lacks rigor. Please explain the J_{sc} difference between the two figures and improve the self-consistency of the analysis.

2. The commercialization of perovskite solar cells is required to withstand moisture ingress and cyclic temperature extremes. In light of these requirements, recent studies have explored the ambient and operational stability (MPP stability) of different encapsulation and adsorption materials in enhancing the stability of perovskite solar cells with more rigorous criteria. (Sci. Adv. 2021, 7: eabi8249; Nat. Energy 2020, 5, 1003–1011) I recommend the author add some stability evaluation from different angles.

3. In Fig. 4e of the manuscript, the author evaluated the stability of encapsulated PSCs with a performance test every 250 hours for the entire duration of 1000 hours. It seems abnormal as compared

to other reported works aimed at preventing lead leakage, where the stability test is usually conducted in less than every 100 hours for each data point. (Sci. Adv. 2021, 7: eabi8249; ACS Energy Lett. 2021, 6, 3443–3449; Adv. Funct. Mater. 2022, 32, 2201036) It's recommended to narrow the interval time between data points of the performance evolution to obtain a more robust and detailed result of the stability of encapsulated PSCs.

4. While the author of this work demonstrated back encapsulation of BCT, several other works highlight the importance of encapsulation from both the front and back sides, as indicated in Supplementary Table 1. The question arises whether there is still a possibility of lead leakage from the front side and cross-sections in extreme scenarios when the devices are cracked into pieces. Therefore, it would be worthwhile to investigate the robustness of the encapsulation technique under such conditions to comprehensively evaluate its efficacy.

5. Suggest improvements. Fig. 1a is not friendly to readers with Arachnophobia, we suggest the author replace the schematic illustration with only spider webs.

6. Compared with some stretchable and self-healable polymer materials for the sequestration of lead, are there any advantages of the BCT materials for further commercialization of PSCs?

Response to Reviewers' comments

Reply: We appreciate the valuable comments and helpful suggestions from the Reviewers. We have considered all the comments carefully. We tried our best to fully answer the questions and comments of the reviewers. The manuscript has been revised according to the helpful suggestions. Point-by-point responses to the Reviewer's questions and comments are listed in the following in detail. All changes have been marked in blue in our revised manuscript.

Response to the comments of Reviewer #1:

This paper proposed a closed-loop Pb management system from adsorption to recycling for Pb in perovskite solar cells by utilizing Pb adsorption materials called bioinspired cage traps. It also simulated the breakdown of solar cells in various real-world environments and showed the results for real-world Pb leakage. These findings are of great significance as the commercialization of perovskite solar cells is becoming more likely. In reviewing this study, there were three main areas. i) Development of BCT Pb adsorbent material and identification of the adsorption mechanism. ii) Simulation of the destruction situation through various real-world simulations. iii) Long-term stability and closed-loop Pb recycling. Although the three areas convey important messages in the solar cell assessment, I believe that the originality and scientific impact in each area are insufficient to satisfy the readership level of Nature Communications. Therefore, I decline publication in this journal. The comments that follow are our rationale.

Reply: We appreciate for the valuable comments and suggestions, which are beneficial to improving our work. We have carefully revised our manuscript according to the proposed suggestions, and further systematically expounded the novelty and scientific impact of this manuscript. All changes have been highlighted in blue in the revised manuscript. We sincerely hope that the revised contents and supplementary experiments will address your concerns.

Our work originally designed and synthesized an eco-friendly biomimetic configuration with synergistic Pb capture functions based on physical adsorption and chemical chelation. In addition, we simultaneously addressed current concerns in the perovskite solar cell (PSCs) domain regarding Pb contamination risk, sustainability of recycling Pb components, and environmental vulnerability of PSCs, which is highly distinctive from the previous materials with limited adsorption capacity and the secondary pollution risk. More importantly, this work provided new insights into establishing objective Pb contamination assessment procedures in the real-world environments, which is a critical issue that has not been given much attention. We believe that this manuscript will be of wide interest to the reader for boosting the commercial feasibility of PSCs.

i) BCT materials. In the description of the BCT adsorbent, similarities with the mesh structure and adhesive properties of spider silk, were mentioned. However, in the case of BCT synthesized based on MM, the porous pores were simply increased from MM

and the chemical reactivity with Pb was given by adsorbing EDA on them. The parent material, MM, is also a commercially available product, and the binding of EDA to Pb utilizes the very common result of Pb binding to an amine group in the organic A site. In other words, it is almost impossible to find anything new respect to materials. When I received this article first, I expected to find a semi-transparent adsorbent with a spider web-like mesh structure, but this was not the case. Neither the structural properties of the material, nor the material itself, make it appropriate to label it bioinspired. We also discussed physical and chemical adsorption, but again, capillarity is proportional to the number of pores, and chemical adsorption does not deviate much from the already known principles. In other words, the results are predictable and do not give the reader any insight. In the synthesis of BCT, it is not clear at what stage and by what reaction the number of pores of MM increases, so there is a lack of in-depth discussion of the specific surface area enhancement, which has a significant impact on Pb adsorption. The results of FT-IR, XPS, etc. are predictable results that should be expected after the completion of BCT synthesis or Pb adsorption, but the manuscript is not complete enough to satisfy the answers, such as materials generation mechanism or give properties.

Reply: We gratefully appreciate the valuable comments. We would like to answer the above concerns point by point.

1) In the description of the BCT adsorbent, similarities with the mesh structure and adhesive properties of spider silk, were mentioned. However, in the case of BCT synthesized based on MM, the porous pores were simply increased from MM and the chemical reactivity with Pb was given by adsorbing EDA on them. The parent material, MM, is also a commercially available product, and the binding of EDA to Pb utilizes the very common result of Pb binding to an amine group in the organic A site. In other words, it is almost impossible to find anything new respect to materials.

Reply: Thank you for providing detailed comments on the novelty aspects of our BCT adsorbent and its potential applications. We would like to elaborate three significant aspects of the material in terms of its structure, adsorption mechanism, and multifunctional characteristics:

1) The novel functional Pb capture molecular structure (CONH-R-NH₂) was synthesized using a new chemical synthesis pathway, inspired by the formation of polypeptide chain micelles of spider's web based on dehydration and condensation reaction. This molecule exhibited the highest binding energy due to the synergistic Pb trapping sites including -CONH and -NH₂ and was implanted into mesoporous matrix to prepare bioinspired "cage traps" (BCT) for efficient Pb adsorption. It is noteworthy that both the ethylenediamine (EDA) and mesoporous matrix (MM) are selected as reactants to initiate this chemical condensation reaction, not solely the final chemical products. In other words, the EDA was not simply adsorbed by MM as the Reviewer's mentioned, because the pivotal molecular structure (CONH-R-NH₂) was produced by designing the new chemical synthesis path (Equation 1).

The bioinspired "cage traps" (BCT) were prepared by implanting CONH-R-NH₂

into the mesoporous matrix. The increased Pb adsorption capacity compared to pure -NH₂ or -COOH molecules is achieved by the CONH-R-NH₂ molecular configuration, which exhibits the highest binding energy as shown in **Fig. 1d** due to the synergistic Pb trapping sites including -CONH and -NH₂. This work strategically unravels a structure-function relation between the molecular configurations and Pb adsorption properties.

2) A new Pb adsorption enhancement mechanism was discovered that involved synergistic Pb trapping functions based on physical adsorption and chemical chelation. This mechanism provided advantages over competitors as it allowed larger spatial reaction chambers to accommodate dissociative Pb ions and improved chemisorption properties. Additionally, the dynamic process of Pb adsorption was analyzed, revealing that the chemical coordination adsorption between CONH-R-NH₂ and Pb ion played a dominant role in immobilizing dissociative Pb ion in the initial state while physical adsorption process occurred later. The adsorption process eventually reached a dynamic equilibrium stage where adsorption and desorption processes achieved equilibrium, ensuring residual Pb concentration did not diminish to zero point. So far, few cases have provided in-depth analysis of the dynamic process upon initializing the physical adsorption and chemical chelation at the same time. However, by understanding the interactions between the adsorbent and adsorbate at a molecular level, our work can improve the design of materials and processes that utilize physical adsorption and chemical chelation. Our work can have a profound influence on the lead management of PSCs, as well as other scientific areas of environmental remediation, catalysis, and drug delivery.

3) BCT exhibited multifunctional characteristics, including Pb adsorption, desorption, recycling, and external stimulus isolation, presenting prominent advantages over reported materials. It is noteworthy that recycle procedures for Pb management of end-of-life PSCs have not been developed. Therefore, this material could have significant potential for use in addressing this issue.

Overall, these findings provide new insights into the design of materials for efficient Pb adsorption, particularly for end-of-life PSCs, and pave the way for further research in this field.

Figure 1d. Graphical representations of the computed adsorption energy model structures and results for COOH, CONH, NH₂ and CONH-R-NH₂.

Revisions in the manuscript:

Line 69-75, Page 3, Line 77-83, Page 4, and Line 93-98, Page 4:

Its natural protein molecular chain structure containing glycine ($\text{NH}_2\text{-CH}_2\text{-COOH}$), alanine ($\text{NH}_2\text{-CH}[\text{CH}_3]\text{-COOH}$), and serine ($\text{NH}_2\text{-CH}[\text{CH}_2\text{OH}]\text{-COOH}$), allowing for strong coordination with the target surface^{17,18}. Inspired by these natural chemical components, we designed a novel functional Pb capture molecular structure (CONH-R-NH₂) based on ethylenediamine (EDA) and implanted it into mesoporous matrices (MM) to synthesize an eco-friendly and low-cost bioinspired “cage traps” (BCT). The BCT is capable of capturing leaked Pb ions from damaged devices, isolating against undesired moisture erosion and recycling Pb from end-of-life devices (Supplementary Table 1). The MM acts as reaction chambers that provide strong physical adsorption processes launched by capillary adsorption effects. This approach achieves synergistic Pb capture effect based on physical and chemical functions. Meanwhile, mathematical modeling was used to elaborate on Pb sorption kinetics based on Langmuir isothermal adsorption equations, where chemisorption induced by strong chelation dominates the Pb adsorption rate. By employing this inexpensive BCT, dissociative Pb ions can be effectively adsorbed up to 99.25%, achieving Pb sequestration efficiency exceeding 99% under extreme weather conditions. More importantly, rigorous and reliable testing procedures are demonstrated by placing PSCs in realistic environments such as Yellow River water and Yellow River soil, providing a benchmark for establishing a systematic and repeatable Pb leakage real-world assessment standard. Moreover, the PSCs maintain 92% of their initial efficiency at 25°C and 50% RH for 1000 h. Sustainable closed-loop Pb management practices have been successfully established, including Pb precipitation, Pb adsorption, Pb desorption and Pb recycling. It is encouraging that the purity of recycled PbI₂ is comparable to commercial 99.99% PbI₂, and it does not compromise the photovoltaic performance of PSCs. This work addresses the issues of capturing leaked Pb ions from in-service PSCs and recycling Pb from end-of-life PSCs simultaneously to alleviate Pb contamination risk, as well as proposing reliable and systematic procedures for assessing hazardous Pb leakage in actual environmental scenarios. The proposed strategy will pave the way for accelerating perovskite photovoltaic technology toward sustainability, eco-friendliness and industrialization.

2) When I received this article first, I expected to find a semi-transparent adsorbent with a spider web-like mesh structure, but this was not the case. Neither the structural properties of the material, nor the material itself, make it appropriate to label it bioinspired.

Reply: Thank you for your comments. We understand that the focus of many published works has been on developing semi-transparent Pb adsorbents for use on the light incident side of PSCs. However, due to the necessary requirement for high optical transparency, the thickness of such adsorbents must be limited, which compromises their Pb adsorption capacity and ambient resistance. Additionally, leaked Pb components tend to flow towards the backside of the device, and effective materials or strategies for capturing them in this location are still limited, as shown in **Figure R1**.

In our work, we have focused on encapsulating environmentally friendly and low-cost Pb capture materials on the backlight side of PSCs. Since material transparency is not a necessary consideration in this location, we have been able to use a Pb capture material with an unlimited thickness requirement. This has resulted in superior Pb capture performance and a steric shield effect under ambient stimuli, which is beneficial for mitigating loss in photovoltaic performance. Thank you again for sharing this insight with us.

Figure R1 (Supplementary Figure. 16 in the revised Supplementary information). The Pb leakage of PSCs encapsulated by semi-transparent Pb adsorbent materials and BCT during actual operation.

The synthetic material labeled as bioinspired can be ascribed to the following three aspects:

1) **Design: the design of Pb capture functional group inspired by adhesive properties of the spider mucilage.** The viscous glycoproteins on the surface of the spider's web behave as viscoelastic solids that can deform like an ideal elastic rubber band, which is essential for retaining the insects trapped in the web long enough to be subdued by the spider [Nat. Commun. **1**, 19 (2010)]. As shown in **Figure R2(I)**, the adhesive properties of spider mucilage, which contain NH₂-CH₂-COOH, NH₂-CH[CH₃]-COOH and NH₂-CH[CH₂OH]-COOH groups that play a crucial role in strength and viscosity due to strong hydrogen bonds, electrostatic attraction, and van der Waals forces [Biol. J. Linn. Soc. **77**, 1–8(2002). J. Exp. Zool. **273**, 186–189(1995)]. These -NH₂ and -COOH groups were selected to adsorb metal Pb ions via chemical coordination, with -COOH groups being grafted onto the mesoporous matrix (MM) through hydrochloric acid activation and ethylenediamine (EDA) molecules endowed with two -NH₂ groups are selected as a reactant to form primary chemical reaction sites.

2) **Functionality: the CONH-R-NH₂ molecular synthetic mechanism of Pb capture agent inspired by the formation mechanism of polypeptide chain micelles in spider web.** The formation mechanism of polypeptide chain micelles in spider webs, where amino-terminal and carboxyl-terminal of non-repetitive regions at both ends of spider silk protein control fiber formation through protein secondary structural

crosslinking induced by dehydration and condensation [Adv. Sci. 9, 2103965(2022)]. To reinforce the Pb capture capability, a novelty functional Pb capture molecular structure (CONH-R-NH₂) was synthesized as a chemical adsorbent via a similar dehydration condensation reaction between the -NH₂ and -COOH for reinforcing the Pb capture capability, as shown in **Figure R2(II)**.

3) **Materials: mesoporous matrixes inspired by reticular and porous structure of spider web.** The delicate spatial structure of spider webs, which provide a large hunting space for prey interception. Mesoporous matrixes with high specific surface area and porous morphology are selected as reaction chambers, where dissociative Pb ions can be confined within the mesoporous cage space, as shown in **Figure R2(III)**.

Overall, the label "bioinspired" here refers to synthetic materials that incorporate these three aspects: design, functionality, and materials. By taking inspiration from nature, we designed a new materials system that is sustainable, efficient, and versatile for closed-loop lead management of perovskite solar cells.

Figure R2(Supplementary Figure. 1 in the revised Supplementary information). Bionic synthesis mechanism of BCT based on spider web hunting mechanism. (I), Selection of Pb capture functional group inspired by adhesive properties of the spider mucilage. (II), Selection of CONH-R-NH₂ molecular synthetic mechanism inspired by the dehydration and condensation reaction of protein chains of spider web. (III), Selection of MM inspired by reticular and porous structure of spider web.

Revisions in the manuscript:

(1) Line 47-53, Page 3:

However, these encapsulation layers cannot prevent Pb component diffusion if the device is broken under external stress due to the limited Pb capture capability. While

the complex preparation process also increases the production costs. Chemical adsorption strategies employing functional chemical materials for external adsorption to minimize Pb leakage have been proposed. Most published works focus on developing semi-transparent Pb adsorbent that can be installed on the light-receiving side of PSCs. High optical transparency is required to avoid decreased photovoltaic performance, limiting the thickness of semi-transparent Pb adsorbent and compromising Pb adsorption capacity and ambient resistance. In practice, the leaked Pb component tends to flow towards the backside of the device due to gravity, and corresponding materials or strategies remain limited. Li et al. integrated a polymer mixture based on cation exchange resin (CER) and ultraviolet (UV) resin into the PSC¹⁰. The Pb component was adsorbed via the rapid cation exchange reaction between abundant sulfonic acid groups (SO_3^-) and Pb^{2+} , achieving 90% Pb sequestration efficiency. Nevertheless, the potential risk of secondary pollution from hazardous solid CER waste emerged as a new environmental issue.

(2) Line 102-108, Page 4 and Line 109-115, Page 5:

Spiders are known to ensnare their prey by spinning a delicate and adhesive web, where the viscous glycoproteins on the surface of the spider's web behave as viscoelastic solids that can deform like an ideal elastic rubber band. It is essential for retaining the insects trapped in the web long enough to be subdued by the spider¹⁹, as shown in Fig. 1a. The adhesive properties of the spider mucilage arise from unique chemical components, including $\text{NH}_2\text{-CH}_2\text{-COOH}$, $\text{NH}_2\text{-CH}[\text{CH}_3]\text{-COOH}$ and $\text{NH}_2\text{-CH}[\text{CH}_2\text{OH}]\text{-COOH}$. The characteristic amino ($-\text{NH}_2$) and carboxyl ($-\text{COOH}$) groups play a crucial role in the strength and viscosity properties due to strong hydrogen bonds, electrostatic attraction and van der Waals force, as shown in Supplementary Fig. 1(I),^{20,21}. Meanwhile, the NH_2 -terminal and COOH -terminal of non-repetitive regions at both ends of spider silk protein control the formation of spider silk fibers and polypeptide chain micelles adhesive through the protein secondary structural crosslinking induced by dehydration and condensation, as shown in Supplementary Fig. 1(II)²². Besides, the $-\text{COOH}$ and $-\text{NH}_2$ groups provide selective complexation with metal ions via strong chemical coordination. Moreover, the unique radial and porous structure of spider web offers huge hunting space and outstanding flexibility (Supplementary Fig. 1(III)). Inspired by these unique features of spiders hunting, we designed a multifunctional mesoporous amino-grafted-carbon net (BCT) to capture dissociative divalent Pb ions and offer a steric shield effect under ambient stimulus (Fig. 1a and Supplementary Fig.1). The MM, analogous to a spider's web, serves as reaction chambers, where the dissociative Pb ions can be confined within mesoporous cage space. $-\text{COOH}$ groups are preliminarily grafted on the MM by hydrochloric acid activation method to form primary chemical reaction sites. Notably, EDA molecules containing two $-\text{NH}_2$ groups were self-assembled onto the activated MM to reinforce the Pb capture capability (analogous to viscous microemulsion on spider's web) (Supplementary Fig. 2).

3) We also discussed physical and chemical adsorption, but again, capillarity is proportional to the number of pores, and chemical adsorption does not deviate much from the already known principles. In other words, the results are predictable and do not give the reader any insight.

Reply: Thanks for the comments. Indeed, from a macroscopic perspective, capillarity

is proportional to the number of pores, while microscopic adsorption processes are determined by gas-liquid-solid three-phase contact properties that dictate the velocity of capillarity. In this work, we found that surface chemical modification technology not only increases the number of mesopores and enlarges the external surface area, but also optimizes interface wettability, which is unpredictable and has never been thoroughly analyzed in literature. In this regard, we systematically analyzed the capillary adsorption theory and found that the variation of the three-phase contact angle plays a significant role in regulating capillary adsorption process.

1) Capillary adsorption analysis. We have deeply discussed the capillary adsorption. Driven by liquid surface tension, the immersed liquid can be spontaneously adsorbed by inner cavities and rise along the contact interface, as shown in **Figure 2b**. The height (h) of the raised liquid level can be calculated by Equation 2 [*J. Fluid. Mech.* **189**, 165-187 (1988).].

$$(\gamma_{SG}-\gamma_{SL})2\pi a\approx\rho g\pi a^2h \quad \text{Equation 2}$$

Figure 2b. Schematic of capillary adsorption.

This implies that the height of the raised liquid level in pipe diameter is proportional to the contact area between liquid and MM or BCT. Combined with Young's Equation: $\cos\theta_c = \frac{\gamma_{SG}-\gamma_{SL}}{\gamma_{LG}}$, the h is further defined as [*Fuel* **234**, 1373-1379 (2018).]:

$$h\approx 2\frac{L_c^2}{a}\cos\theta_c \quad \text{Equation 3}$$

Where, γ_{SG} , γ_{SL} , γ_{LG} are the surface tension coefficients between solid-gas and solid-liquid, liquid-gas, respectively, a is the capillary radius, ρ is the liquid density, g is the acceleration of gravity, h is the height of the liquid level rise, θ_c is the contact angle, $L_c = \sqrt{\frac{\gamma_{LG}}{\rho g}}$.

The theoretical result of liquid level raised height is $h > 0$ when the contact angle between the solution and the adsorbate is an acute. And the smaller the contact angle, the larger the h value, indicating that capillary adsorption is more likely to occur. Therefore, we further verified the raised liquid level of MM and BCT via testing the wettability of the deionized water (DI water) on the MM and BCT surfaces. As shown in **Supplementary Figure 8**, BCT exhibits better wettability with a lower contact angle

(38.5°) compared with the MM with 58.9° contact angle, indicating the BCT is more conducive for capillary adsorption.

Supplementary Figure 8. The corresponding water contact angles of MM (58.9°) and BCT (38.5°) surface. BCT has better wettability than MM, which is more conducive for capillary adsorption

2) Chemical Pb adsorption analysis. We propose a novel Pb capture molecular structure (CONH-R-NH₂) based on the formation mechanism of amino acid peptide chains, as shown in **Figure R3**. The molecular configuration affects the Pb adsorption capacity by calculating the binding energies between different chemical groups and Pb ions. As discussed above, the formation of an inner cavity mesoporous structure with more cavities is beneficial to confining the dissociative Pb ion within mesoporous cage space, yielding excellent capillary physical adsorption capacity. More importantly, the chemical molecular grafted on a mesoporous matrix play an important role in regulating interface wettability, providing the new insight for comprehensively understanding Pb adsorption mechanism of BCT.

Figure R3 (Supplementary Figure. 4 in the revised Supplementary information). The design idea of Pb capture molecular structure (CONH-R-NH₂).

3) Dynamic Pb ion adsorption analysis. We discovered a dynamic Pb ion adsorption process. As shown in **Figure 2e**, we further divided the whole adsorption curves of BCT and MM into three adsorption reaction processes, including: (I) chemical

dominated adsorption stage (0~5 min), (II) physical dominated adsorption stage (5~120 min) and (III) equilibrium adsorption stage (120~180 min). Sufficient chemical adsorption sites on the surface play a pivotal role in immobilizing the dissociative Pb ion in the initial state. The adsorption rate defers when the chemical active sites of the adsorbent surface approach saturation, moving into the physical dominated stage where capillary adsorption plays a dominant role. In the dynamic equilibrium adsorption stage, the residual Pb concentration does not diminish to zero point as the adsorption and desorption process of Pb ions achieve equilibrium. The Pb equilibrium adsorption capacity (q_e) of MM and BCT was quantitatively determined as 14.220 and 32.934 mg/g, respectively.

Figure 2e. Process analysis of MM and BCT reacting with PbI_2 .

In general, an inner cavity mesoporous structure with more cavities is beneficial to confine the dissociative Pb ion within mesoporous cage space, yielding excellent capillary physical adsorption capacity. Moreover, the chemical molecular grafted on a mesoporous matrix plays a crucial role in regulating interface wettability, providing new insight for comprehensively understanding the Pb adsorption mechanism of BCT.

Revisions in the manuscript:

(1) Line 189-191, Page 7:

Interestingly, $-CONH-R-NH_2$ exhibits supreme binding energies of 3.3 eV, due to the coupling molecular configuration containing multiple Pb binding sites such as $-NH_2$ and $-CONH$, achieving synergistic adsorption of Pb ions (Supplementary Fig. 4). According to the results, it can be inferred that a significant chemical interaction exists between the BCT group and PbI_2 . The chemical adsorption capacity of BCT for Pb ion has been greatly improved via the step-by-step process of hydrochloric acid activation and EDA molecular self-assembly.

(2) Line 224-229, Page 8 and Line 230-242, Page 9:

The porous media is endowed with elongated inner cavities that can be regarded as capillaries for triggering the capillary adsorption action²⁵. Driven by liquid surface tension, the immersed liquid can be spontaneously adsorbed by inner cavities and rise

along the contact interface, as shown in Fig. 2b. The height (h) of the raised liquid level can be calculated by Equation 2²⁹.

$$(\gamma_{SG}-\gamma_{SL})2\pi a\approx\rho g\pi a^2h \quad \text{Equation 2}$$

It can be concluded that the height of the raised liquid level in pipe diameter is proportional to the contact area between liquid and MM or BCT. Combined with Young's Equation: $\cos\theta_c=\frac{\gamma_{SG}-\gamma_{SL}}{\gamma_{LG}}$, h is further defined as³⁰:

$$h\approx 2\frac{L_c^2}{a}\cos\theta_c \quad \text{Equation 3}$$

Where, γ_{SG} , γ_{SL} , γ_{LG} are the surface tension coefficients between solid-gas and solid-liquid, liquid-gas, respectively; a is the capillary radius; ρ is the liquid density; g is the acceleration of gravity; h is the height of the liquid level rise; θ_c is the contact angle; $L_c=\sqrt{\frac{\gamma_{LG}}{\rho g}}$.

It can be seen that the theoretical result of liquid level raised height is $h > 0$ when the contact angle between the solution and the adsorbate is an acute. The smaller the contact angle, the larger the h value, indicating that capillary adsorption is more likely to occur. Therefore, we further verified the raised liquid level of MM and BCT via testing the wettability of the deionized water (DI water) on the MM and BCT surfaces. As shown in Supplementary Fig. 8, BCT exhibits better wettability with a lower contact angle (38.5°) compared with the MM with 58.9° contact angle, indicating the BCT is more conducive for capillary adsorption. Therefore, a synergistic capture effect based on strong physical adsorption and chemical chelation can be realized.

4) In the synthesis of BCT, it is not clear at what stage and by what reaction the number of pores of MM increases, so there is a lack of in-depth discussion of the specific surface area enhancement, which has a significant impact on Pb adsorption.

Reply: Thanks for your comments. In this work, we synthesized BCT using a three-stage process: (I) Initial state, (II) Ultrasound-assisted structural separation stage and (III) Surface chemical reconfiguration stage (**Figure R4A(I)**). According to the suggestion, we used SEM to observe the surface morphology of BCT in these three stages and determine at what stage and by what reaction the number of pores of MM increases. As shown in **Figure R4a(II)** and **Figure R4b, in stage I**, pristine MM shows a relatively smooth surface with an estimated 13 pores per $100 \mu\text{m}^2$. **In stage II**, MM was treated with high-frequency ultrasonic vibration for over 90 mins to reduce the particle size, resulting in better dispersion in solution. As shown in **Figure R4b**, adjacent pores fused together and new pores appeared. The number of pores increased to an estimated 36 per $100 \mu\text{m}^2$ and the average pore diameter became larger. **In stage III**, EDA molecular was grafted on the MM to reconstruct the surface chemical state with the assistance of high-frequency ultrasonic vibration. This process produced the new CONH-R-NH₂ molecular configuration and created a rougher surface with more pores appearing (**Figure R4a**). The number of the pores increased to approximately 52 per $100\mu\text{m}^2$. The external surface area of BCT is quantitatively evaluated as $108.4398 \text{ m}^2\text{g}^{-1}$ via Brunauer–Emmett–Teller surface area and pore size analyzer, which is more

than twice as large as MM (Supplementary Fig. 6). The detailed open cavities of BCT were further visualized by 3DOP (Fig. 2a and Supplementary Fig. 7). The average surface roughness of BCT increases from 22.393 μm to 32.535 μm after grafting EDA due to generating a large number of new pores. The high-frequency ultrasonic vibration during the EDA grafting process physically reconstructed the surface structure to some extent due to the material's potential plasticity. These enhancements of specific surface area and pore characteristics have significant impacts on Pb adsorption by BCT.

Figure R4 (Supplementary Figure. 5 in the revised Supplementary information).
a, (I) (II) The SEM pictures of different stages in BCT synthesis route. b, The number of pores in different stages of BCT synthesis route.

Revisions in the manuscript:

Line 194-202, Page 7 and Line 203-212, Page 8:

The macroscopic morphology of the MM surface is observed to change in a large-scale coupled with the EDA self-assembly grafting process, which is confirmed by scanning electron microscopy (SEM), three-dimensional optical profilometer (3DOP), and high-pressure chemisorption. The synthesis of BCT can be divided into three stages, including (I) Initial state, (II) Ultrasound-assisted structural separation stage and (III) Surface chemical reconfiguration stage (Supplementary Fig. 5a(I)). The surface morphology of BCT in these three different stages was observed via SEM to figure out what stage and by what reaction the number of pores of MM increases. As shown in Supplementary Fig. 5a(II) and Supplementary Fig. 5b, in the stage I, the pristine MM shows a relatively smooth surface and the number of pores was estimated around 13 per $100 \mu\text{m}^2$. In the stage II, MM was treated by high-frequency ultrasonic vibration over 90 min to reduce the particle size, providing better dispersion in solution. It can be clearly seen that the adjacent pores are fused together and new pores appear, with an estimated number of pores around 36 per $100 \mu\text{m}^2$ and the larger average pore diameter (Supplementary Fig. 5b). In stage III, EDA molecular is grafted on the MM to reconstruct the surface chemical state with the assistance of high-frequency ultrasonic vibration, producing the new CONH-R-NH₂ molecular configuration. The synthesized BCT surface presents a rougher surface with more pores appearing (Supplementary Fig. 5a). The number of the pores is around 52 per $100 \mu\text{m}^2$. The external surface area of BCT is quantitatively evaluated as $108.4398 \text{ m}^2\text{g}^{-1}$ via Brunauer–Emmett–Teller surface area and pore size analyzer, which is more than twice as large as MM (Supplementary Fig. 6). The detailed open cavities of BCT were further visualized by 3DOP, as shown in Fig. 2a and Supplementary Fig. 7. The average surface roughness of BCT increases from $22.393 \mu\text{m}$ to $32.535 \mu\text{m}$ after grafting EDA due to generating a large number of new pores. It is reasoned that the high-frequency ultrasonic vibration during the EDA self-assembly grafting process physically reconstructs the surface structure to some extent due to the potential material plasticity. The porous media is endowed with elongated inner cavities that can be regarded as capillaries for triggering the capillary adsorption action²⁵.

5) The results of FT-IR, XPS, etc. are predictable results that should be expected after the completion of BCT synthesis or Pb adsorption, but the manuscript is not complete enough to satisfy the answers, such as materials generation mechanism or give properties.

Reply: Thanks for your kind suggestions. We have added detailed mechanistic analysis regarding FT-IR, XPS, etc. to the manuscript in blue to provide a more complete understanding of the materials generation mechanism or properties.

Chemical synthesis design of the BCT. As shown in **Figure 1b**, the FTIR spectrum of the MM shows characteristic peaks at around 3446 cm^{-1} and 1630 cm^{-1} , which are assigned to the stretching vibration of the hydroxyl group (OH) and carbonyl group (C=O) of the -COOH group respectively, confirming the presence of -COOH in the mesoporous matrix induced by the hydrochloric acid activation treatment. The new peaks regarding N-H tensile vibration at around 3417 cm^{-1} and the out-of-plane bending

vibration absorption of -NH_2 at around 880 cm^{-1} can be clearly observed in BCT, indicating the terminal -NH_2 group from EDA molecular has been successfully installed in BCT. Notably, the characteristic strong vibration absorption peaks that appear at 1633 cm^{-1} and 1579 cm^{-1} are related to the stretching vibration of the -C=O and deformation vibration of -N-H of the acrylamide group (-CONH), which is mainly stemming from the formation of $p\text{-}\pi$ conjugation between the carbonyl groups and the lone-pair electron of nitrogen atoms on amide groups [Acc. Chem. Res. **41**, 432(2008)]. Compared with the peak exists at 1630 cm^{-1} which belongs to -C=O group in -COOH of MM, the peak of -C=O group in BCT shifts to 1633 cm^{-1} , evidently demonstrating that the charge distribution around -C=O has changed and -COOH group participated in the self-assembly grafting process of EDA. Besides, the strong peaks at 1453 cm^{-1} and 1099 cm^{-1} are attributed to the C-N antisymmetrical stretching vibration in -CONH and $\text{-CH}_2\text{-NH}_2$, respectively. Moreover, the presence of substate signals of N1s at 395.8 eV and 396.9 eV in X-ray photoelectron spectroscopy (XPS) also prove that the -NH_2 and -CONH based groups exist on the BCT surface, as shown in **Figure 1c** [Carbohydr. Polym. **234**, 115911 (2020)]. Given the spontaneous dehydration condensation reaction between the -NH_2 and -COOH terminal for amidating polypeptide chain in spider silk protein, it is confirmed that the EDA molecules is grafted onto the BCT surface accompanied by the reconstruction of surface states to produce the new functional groups -CONH-R-NH_2 via the dehydration condensation between the -COOH and -NH_2 groups, which is triggered by the assistance of ultrasonic waves.^[24] On the basis of the analysis of the molecular structure evolution, the chemical synthesis routes of the BCT can be deduced as Equation 1.

Pb capturing mechanism based on the chemical chelation. In order to unravel the Pb capturing mechanism based on the chemical chelation between the functional groups that grafted on the BCT surface and Pb ions, the detailed deviation of the signal peaks in XPS and FTIR are analyzed. Figure 1c and d shows the XPS peak spectra of N and Pb elements of pure PbI_2 and PbI_2 mixed with BCT powders (referred as $\text{PbI}_2\text{@BCT}$ samples). The peaks corresponding to $\text{Pb } 4f_{5/2}$ and $\text{Pb } 4f_{7/2}$ shift from 139.6 eV and 134.7 eV to higher binding energies of 140.6 eV and 135.8 eV respectively for $\text{PbI}_2\text{@BCT}$ samples, implying inner shell electrons of Pb ions possess with enhanced binding energy, which is closely associated with the bonding state. Furthermore, the substate N signals in -NH_2 peak of 395.8 eV and the -CONH peak of 396.9 eV on the BCT surface are remarkably upward to 396.6 eV and 397.9 eV upon encountering Pb ions. Besides, in the Figure 1b, it is found the peak positions of NH and -C=O in the -CONH group of $\text{PbI}_2\text{@BCT}$ samples obviously downward shift from 3417 cm^{-1} and 1630 cm^{-1} to 3401 cm^{-1} and 1600 cm^{-1} in FTIR spectra, respectively. These results provide strong evidence regarding the chemical coordination interaction between the -CONH and Pb ion. -NH and -C=O as Lewis base adduct in -CONH group with electron-donating properties offer delocalized electrons, thus altering the surrounding chemical environment and inducing peripheral coulomb interactions with positive Pb ions [Science **366**, 1509(2019)]. Interestingly, out-of-plane deformation vibration absorption peak at 880 cm^{-1} assigning to the terminal -NH_2 group is

disappeared while accompanied by appearing new telescopic vibration peak of -NH_2^+ at 2315 cm^{-1} , which due to the delocalized lone pair of electrons on the nitrogen atom shift to the positively charged Pb^{2+} to form metal coordination complex [J. Phys. Chem. B 113, 15914-15920 (2009)].

Figure 1. **b**, FTIR spectra of MM, BCT and PbI₂@BCT. **c**, XPS spectra of N1s for MM, BCT and PbI₂@BCT. **d**, Graphical representations of the computed adsorption energy model structures and results for COOH, CONH, NH₂ and CONH-R-NH₂.

For an in-depth understanding of how the molecular configuration influences the chelating strength, first-principles density-functional theory (DFT) was conducted to calculate binding energies between different chemical groups and Pb ions. As depicted in **Figure 1d**, compared with the original -COOH (3.17 eV) and -NH₂ (2.87 eV) groups, -CONH as one of the reaction products shows a binding energy of 3.04 eV benefiting from the -C=O and -NH in the -CONH exhibits Lewis basicity in which lone pair electrons in oxygen atom and nitrogen atom both delocalized to form the basis of the strong interaction [Science 366, 1509(2019). Coord. Chem. Rev. 434, 213809(2021)] Interestingly, -CONH-R-NH₂ exhibits supreme binding energies of 3.3 eV, as a result of the coupling molecular configuration containing multiple Pb binding sites such as NH₂ and CONH achieve synergistic adsorption of Pb ions. According to the above results, it is inferred that there exists a significant chemical interaction between the group of BCT and PbI₂. The chemical adsorption capacity of BCT for Pb ion has been greatly improved via the step-by-step of hydrochloric acid activation and EDA molecular self-assembly.

In summary, the addition of detailed mechanistic analysis regarding FT-IR, XPS, etc. helps provide a better understanding of the material generation mechanism and

properties, as well as the Pb capturing mechanism based on chemical chelation between functional groups grafted on the BCT surface and Pb ions.

Revisions in the manuscript:

Line 135-136, Page 5, Line 138-141, Page 5, Line 142-160, Page 6, Line 172-174, Page 6, Line 180-186, Page 7 and Line 189-192, Page 7:

As shown in Fig. 1b, the Fourier transform infrared spectroscopy (FTIR) of the MM shows characteristic peaks at around 3446 cm^{-1} and 1630 cm^{-1} assigned to the stretching vibration of the hydroxyl group (-OH) and carbonyl group (-C=O) of the -COOH group respectively, confirming the presence of -COOH in the mesoporous matrix induced by the hydrochloric acid activation treatment. Compared to MM, the BCT demonstrates new peaks regarding -N-H tensile vibration at around 3417 cm^{-1} and the out-of-plane deformation vibration absorption of -NH₂ at around 880 cm^{-1} , indicating the terminal -NH₂ group from EDA molecular has been successfully grafted onto the BCT. Additionally, the strong vibration absorption peaks at 1633 cm^{-1} and 1579 cm^{-1} are related to the stretching vibration of the -C=O and deformation vibration of N-H of the acylamide group (-CONH), which indicates that p- π conjugation between the carbonyl groups and the lone-pair electron of nitrogen atoms on amide groups has occurred²³. Compared with the peak at 1630 cm^{-1} belonging to -C=O group in -COOH of MM, the peak of the -C=O group in BCT shifts to 1633 cm^{-1} , demonstrating that the charge distribution around -C=O has changed and -COOH group has participated in the self-assembly grafting process of EDA. The strong peak at 1099 cm^{-1} is attributed to the C-N antisymmetrical stretching vibration. Moreover, the X-ray photoelectron spectroscopy (XPS) analysis indicates the presence of substate signals of N1s at 395.8 eV and 396.9eV, proving that the -NH₂ and CONH-based groups exist on the BCT surface, as shown in Fig. 1c²⁴. Given the spontaneous dehydration condensation reaction between the -NH₂ and -COOH terminal for amidating polypeptide chain in spider silk protein, it is confirmed that the EDA molecules are grafted onto the BCT surface. This reconstruction of surface states produces new functional groups (-CONH-R-NH₂) via the dehydration and condensation between the -COOH and -NH₂ groups, triggered by the assistance of ultrasonic waves²⁵. Based on the analysis of the molecular structure evolution, the chemical synthesis route of the BCT can be deduced as Equation 1.

Pb capturing mechanism based on the chemical chelation

In order to understand the Pb capturing mechanism based on the chemical chelation between the functional groups that grafted on the BCT surface and Pb ions, the deviation of the signal peaks in XPS and FTIR are analyzed. Fig. 1c and Supplementary Fig. 3 show the XPS peak spectra of N and Pb elements of pure PbI₂ and PbI₂ mixed with BCT powders (referred as PbI₂@BCT samples). The peaks corresponding to Pb 4f_{5/2} and Pb 4f_{7/2} shift from 139.6eV and 134.7eV to higher binding energies of 140.6 eV and 135.8 eV for PbI₂@BCT samples, respectively. This indicates that the inner shell electrons of Pb ions possess enhanced binding energy due to the bonding state. Furthermore, the substate N signals in -NH₂ peak of 395.8 eV and the -CONH peak of 396.9 eV on the BCT surface are remarkably upward to 396.6 eV and 397.9 eV upon encountering Pb ions. Besides, in the Fig. 1b, the peak positions of -NH and -C=O in the -CONH group of PbI₂@BCT samples obviously downward shift from 3417 cm^{-1} and 1630 cm^{-1} to 3401 cm^{-1} and 1600 cm^{-1} in FTIR spectra, respectively. These results

provide strong evidence regarding the chemical chelation between the -CONH and Pb ion. -NH and -C=O as Lewis base adducts in -CONH group with electron-donating properties offer delocalized electrons, thus altering the surrounding chemical environment and inducing peripheral coulomb interactions with positive Pb ions²⁶. Interestingly, out-of-plane deformation vibration absorption peak at 880 cm⁻¹, assigning to the terminal -NH₂ group, disappears while accompanied by appearing new telescopic vibration peak of -NH₂⁺ at 2315 cm⁻¹. This is due to the delocalized lone pair of electrons on the nitrogen atom shifting to the positively charged Pb²⁺ to form metal coordination complex²⁷.

To understand how the molecular configuration influences the chelating strength, first-principles density-functional theory (DFT) was conducted to calculate binding energies between different chemical groups and Pb ions. As depicted in Fig. 1d, compared with the original -COOH (3.17 eV) and -NH₂ (2.87 eV) groups, -CONH as one of the reaction products shows a binding energy of 3.04 eV. This is because the -C=O and -NH in the -CONH exhibit Lewis basicity where lone pair electrons in oxygen and nitrogen atoms delocalized to form the basis of the strong interaction^{26, 28}. Interestingly, -CONH-R-NH₂ exhibits supreme binding energies of 3.3 eV, due to the coupling molecular configuration containing multiple Pb binding sites such as -NH₂ and -CONH, achieving synergistic adsorption of Pb ions (Supplementary Fig. 4). According to the results, it can be inferred that a significant chemical interaction exists between the BCT group and PbI₂. The chemical adsorption capacity of BCT for Pb ion has been greatly improved via the step-by-step process of hydrochloric acid activation and EDA molecular self-assembly.

ii) Simulation of destruction situation of PVSK solar cells. This is the part where we can give the most originality to this research. However, there is a structural limitation in the encapsulation of BCTs by mixing them with conductive carbon. If the destruction only occurred on the light-accepting glass side, the encapsulated BCTs would not adsorb Pb enough. To solve this problem, it is thought that if double-sided application is used, there will be a sharp decrease in efficiency due to the black color of the carbon.

Reply: Thank you for this constructive comment. Our work aims to provide a complementary strategy from a new perspective in practical application scenarios. In practice, double-sided application is feasible by employing a transparent lead adsorption material on the light-accepting glass side and simultaneously packaging a BCT on the back side, providing a synergistic effect to suppress lead leakage. Most works focus on developing functional materials for use on the light-accepting glass side. However, leaked Pb components tend to flow to the back side along the cracks within the damaged PSCs by gravity, because modules are installed at a large inclination angle to optimize sunlight exposure for that location, as shown in **Figure R5**. Therefore, it is essential to develop a new strategy for backside packaging, which is crucial to ensure reliable protection in practical applications.

Schematic diagram of lead leakage in practical application of perovskite solar cells

Figure R5 (Supplementary Fig. 13 in the revised Supplementary information). The Pb leakage process of glass-encapsulated PSCs in actual operation.

In our work, we have performed hail simulations to evaluate the Pb leakage sequestration efficiency of broken BG-PSCs in heavy rainfall flushing, acid rain, Yellow River water, and Yellow River soil, as shown in **Figure R6**. Note that, the destruction occurred only on the light-accepting glass side. It is demonstrated that the BCT exhibits superior Pb capture performance and a steric shield effect under ambient stimulus when only the light-accepting glass was damaged. The damaged PSCs encapsulated with BCT exhibit 99.86% and 91.68% of Pb sequestration efficiency in the simulation of heavy rainfall flushing and acid rain, respectively. And the pollution of end-of-life PSCs to the Yellow River soil was effectively limited at non-contamination levels (91.60% of Pb sequestration efficiency).

Light-accepting side broken

Figure R6 (Supplementary Fig. 12 in the revised Supplementary information). Typical photographs of damaged U-PSC, G-PSC and BG-PSC for Pb leakage tests.

Besides, we conducted additional experiments to compare the effect of different install position of functional materials on Pb leakage suppression performance under extreme destructive conditions. First, the mature acrylate ultraviolet curing resin was selected to package on light-accepting sides of PSCs [*Nat. Energy* **4**, 585–593 (2019).], which have been demonstrated excellent capability of Pb leakage suppression in the previous works, and the BCT was installed on the back side of PSCs (BG-PSC) for

comparison (**Figure R7a**). The unpackaged PSC (U-PSC), glass encapsulated PSC (G-PSC), resin encapsulated PSC (R-PSC) and BG-PSC were completely cracked into pieces via violent hail attacking simulation (**Figure R7b**). Then, we immersed them in DI water for 6h to access the concentration of leaked Pb ions. As shown in **Figure R8**, the leaked Pb ions are 7.84, 5.57 and 4.03 ppm for U-PSC, G-PSC and R-PSC, respectively. Encouragingly, only 0.876 ppm of Pb ions leak from BG-PSC. Compared with U-PSC, G-PSC and R-PSC, which exhibit the Pb sequestration efficiency of only 0.00%, 26.79% and 48.59%, respectively, BG-PSC exhibits Pb sequestration efficiency as high as 88.83%. Therefore, it is concluded that BCT encapsulation strategy displays higher Pb sequestration efficiency compared with conventional encapsulation strategies that pack the material on light- accepting glass side, even when the devices are cracked into pieces. It is mainly due to the advantage of large adsorption capacity and fast adsorption rate for BCT materials.

Furthermore, the significance of this work is to propose BCT packaging strategy based on environmentally friendly and low-cost BCT materials. The thickness of such material is not limited by the rules of optical transmission, thus ensuring superior Pb adsorption capacity and Pb sequestration efficiency. Moreover, the Pb ions captured by the BCT layer can be recycled and reused via the Pb closed-loop management strategy, which significantly increases the environmental friendliness of PSCs and reduces their cost of commercial application. In practical applications, BCT materials can be an alternative to cooperating with translucent Pb capture materials for double-sided application to achieve synergistic encapsulation for enhanced Pb leakage suppression.

Figure R7 (Supplementary Figure. 14 in the revised Supplementary information).

a, The methods of encapsulating PSCs with resin and BCT. b, Typical photographs of completely cracked U-PSC, G-PSC, R-PSC and BG-PSC.

Figure R8 (Supplementary Figure. 15 in the revised Supplementary information). The Pb leakage and Pb sequestration efficiencies of the completely cracked U-PSC, G-PSC, R-PSC and BG-PSC in DI water.

Revisions in the manuscript:

Line 340-353, Page 12 and Line 354-363, Page 13:

When PSCs are subjected to external impacts such as hail during practical application, the leaked Pb components also tend to flow to the backside along the cracks within the damaged PSCs by gravity due to the modules being installed at a large inclination angle to optimize sunlight exposure for that location (Supplementary Fig. 13). Besides, we conducted additional experiments to compare the effect of different installation positions of functional materials on Pb leakage suppression performance under extreme destructive conditions. First, the mature acrylate ultraviolet curing resin was selected to package on light- accepting sides of PSCs, which have demonstrated with excellent capability of Pb leakage suppression in the previous works, and the BCT was installed on the backside of PSCs (BG-PSC) for comparison (Supplementary Fig. 14a). The unpackaged perovskite solar cell (U-PSC), glass-encapsulated perovskite solar cell (G-PSC), resin encapsulated PSC (R-PSC) and BG-PSC were completely cracked into pieces via violent hail attacking simulation (Supplementary Fig. 14b). Then, we immersed them in DI water for six hours and measured the concentration of leaked Pb ions. As shown in Supplementary Fig. 15, the leaked Pb ion concentrations were 7.84, 5.57 and 4.03 ppm for U-PSC, G-PSC and R-PSC, respectively.

Encouragingly, only 0.876 ppm of Pb ions leaked from BG-PSC. Compared with U-PSC, G-PSC and R-PSC, which exhibit the Pb sequestration efficiency of only 0.00%, 26.79% and 48.59%, respectively, BG-PSC exhibits Pb sequestration efficiency as high as 88.83%. Therefore, it is concluded that the BCT encapsulation strategy displays much higher Pb sequestration efficiency compared with the conventional encapsulation strategy of packing the material on light-accepting glass side, even when the devices are cracked into pieces (Supplementary Fig. 16). This is mainly attributed to the advantage of large adsorption capacity and fast adsorption rate for BCT materials.

iii) Recycling

This part is almost impossible to give the value of originality, scientific discovery, etc. to be published in Nature Communications because its mechanism is a well established already, and demonstrated similarly so many times.

Reply: Thanks for your comments. In response to your comment about the difficulty in evaluating the originality and scientific discovery of our manuscript, we have included **Supplementary Table 1** to demonstrate that most of the reported works have mainly focused on developing function materials to suppress the Pb leakage and have not established a sustainable closed-loop Pb management engineering that includes the Pb recovery process. Our work successfully establishes this sustainable closed-loop process via four critical steps: Pb precipitation, Pb adsorption, Pb desorption and Pb recycling. The bonding mode of CONH-R-NH₂ to Pb ions and the unique structure and versatility of the BCT are key contributors to our success.

In addition, Pb adsorption and Pb desorption steps in our work differ from that of Chen et al. [*Nat. Commun.* **12**, 5859 (2021).] in that they used a carboxylic acid cation-exchange resin as adsorbent to ensure Pb adsorption efficiency based on the ionic bonding between carboxylic acid and Pb ion, as shown in **Equation 4**. Besides, they achieved high Pb release based on hydrogen ionic exchange reaction between strong acid (HNO₃) and weak acid (-COOH). In contrast, our Pb ions were captured via the chemical coordination bonding between Pb ions and RCO-NH-R, RNH₂, CONH-R-NH₂, as shown in **Equation 6-8** and **Figure R9a**. Our Pb release process was mainly dominated by secondary protonation reaction, which is different from the “Strong acid to make weak acid” reaction process.

In that of Chen et al. [*Nat. Commun.* **12**, 5859 (2021).], the ionic reactions in solution will always shift towards a state where the concentration gradient will decrease. As shown in **Equation 5**, H⁺ ionized from HNO₃ is prone to bond with COO⁻ due to the different acid dissociation constant (K_a(HNO₃)>K_a(RCOOH)).

In our work, the Pb ions were captured via the chemical coordination bonding between Pb ions and RCO-NH-R, RNH₂, CONH-R-NH₂, as shown in **Equation 6-8** and **Figure R9a**. The Pb release process was mainly dominated by secondary protonation reaction, which is distinct from the “Strong acid to make weak acid”

reaction process.

Figure R9 (Supplementary Figure. 27 in the revised Supplementary information). a, Pb ions capture mechanism of RCO-NH-R, RNH₂ and CONH-R-NH₂. b, Protonation mechanism of RCO-NH-R, RNH₂ and CONH-R-NH₂. c, Protonated structure of RCO-NH-R, RNH₂ and CONH-R-NH₂.

In a highly acidic solution environment, the protonated hydrogen ion could be chelated by the lone pair around N element in RCO-NH-R, RNH₂ and CONH-R-NH₂, forming RCO-NH₂⁺-R, RNH₃⁺, and CONH₂⁺-R-NH₃⁺. The secondary protonation reaction destabilized the existed binding force between the RCO-NH-CH₂-CH₂-NH₂ and Pb ions because H processes larger electronegativity than Pb (H (2.2)>Pb (1.9)) [*Chem. Eng. J.* **197**, 88-100 (2012).], resulting in releasing the captured Pb ions as shown in Equation 9-11 and Figure R9b and R9c.

Here, we have demonstrated the protonation process via characterizing the FTIR of the mixed solution of CONH-R-NH₂ with PbI₂ (referred as CONH-R-NH₂@PbI₂) and the mixed solution of CONH-R-NH₂, PbI₂ with HNO₃ (referred as CONH-R-NH₂@PbI₂+HNO₃), respectively. As shown in **Figure R10**, CONH-R-NH₂@PbI₂ shows characteristic peaks at around 3340 cm⁻¹ and 3277 cm⁻¹, which are assigned to the stretching vibration of -NH in -NH₂. And the characteristic strong vibration absorption peak that appear at 1550 cm⁻¹ is related to the stretching vibration of the -C=O in the amide group(-CONH). In comparison with CONH-R-NH₂@PbI₂, it is found the peak positions of the stretching vibration of -NH in CONH-R-NH₂@PbI₂+HNO₃ samples obviously shift from 3340 cm⁻¹ and 3277 cm⁻¹ to 3228 cm⁻¹ and 3165 cm⁻¹ in FTIR spectra, respectively. The new peaks regarding the stretching vibration of protonated amine groups (-NH₃⁺) at around 2823 cm⁻¹ and 2596 cm⁻¹ can be clearly observed in CONH-R-NH₂@PbI₂+HNO₃, indicating that the -NH₂ group was protonated via the H⁺ ionized by HNO₃ [*Nat. Commun.* **12**, 6856 (2021)]. Meanwhile, the new peak appears at around 2535 cm⁻¹ and 1514 cm⁻¹, which are assigned to the stretching vibration of protonated amide(-NH₂⁺) and protonated amide II, manifesting the -CONH was protonated by H⁺ [*Bioresource Technol.* **268**, 454–459 (2018). *J. Environ. Chem. Eng.* **9**, 104700(2021)]. Besides, the peak positions of the stretching vibration of -C=O also obviously shifts from 1550 cm⁻¹ to 1606 cm⁻¹. The reason is attributed to that H⁺ ionized by HNO₃ coordinates with the lone pair on the N atom on the -CONH-, thus breaking the conjugated system of the -CONH and making the C=O bond shift to high frequency due to its enhanced double bond properties [*J. Am. Chem. Soc.* **127**, 4445–4453 (2005). *Appl. Surf. Sci.* **444**, 387–398(2018)]. The above results show that the Pb desorption is achieved based on the protonation of CONH-R-NH₂ molecular configuration by HNO₃. Therefore, in this work, the adsorption and desorption mechanisms of Pb in the steps of Pb adsorption and desorption are different from those of the previous literature.

We hope these explanations clarify the originality and scientific discovery of our manuscript and address your concerns. We have provided additional details and evidence in our revised manuscript and supplementary information to support our claims.

Figure R10 (Supplementary Figure. 28 in the revised Supplementary information). FTIR spectra of CONH-R-NH₂@ PbI₂ and CONH-R-NH₂@ PbI₂+ HNO₃

Revisions in the manuscript:

Line 479-500, Page 17 and Line 501-513, Page 18:

In procedures I-II, the BCT can effectively adsorb Pb ions from the solution with an adsorption rate of 99.77% via chemical coordination bonding between Pb ions and RCO-NH-R, RNH₂, CONH-R-NH₂, as shown in Supplementary Fig. 27a. In procedure III, the Pb release process was mainly dominated by secondary protonation reaction. In a highly HNO₃ solution environment, the protonated hydrogen ion could be chelated by the lone pair around N element in RCO-NH-R, RNH₂ and CONH-R-NH₂, forming RCO-NH₂⁺-R, RNH₃⁺, and CONH₂⁺-R-NH₃⁺. The secondary protonation reaction destabilized the existed binding force between the RCO-NH-CH₂-CH₂-NH₂ and Pb ions because H processes larger electronegativity than Pb (H (2.2)>Pb (1.9))³⁷, resulting in releasing the captured Pb ions as shown in Equation 4-6 and Supplementary Fig. 27b and c.

We have demonstrated the protonation process via characterizing the FTIR of the mixed solution of CONH-R-NH₂ with PbI₂ (referred as CONH-R-NH₂@PbI₂) and the mixed solution of CONH-R-NH₂, PbI₂ with HNO₃ (referred as CONH-R-NH₂@PbI₂+HNO₃), respectively. As shown in Supplementary Fig. 28, CONH-R-NH₂@PbI₂ shows characteristic peaks at around 3340 cm⁻¹ and 3277 cm⁻¹, which are assigned to the stretching vibration of -NH in -NH₂. And the characteristic strong

vibration absorption peak that appear at 1550 cm^{-1} is related to the stretching vibration of the -C=O in the -CONH group. In comparison with $\text{CONH-R-NH}_2@\text{PbI}_2$, it is found the peak positions of the stretching vibration of -NH in $\text{CONH-R-NH}_2@\text{PbI}_2+\text{HNO}_3$ samples obviously shift from 3340 cm^{-1} and 3277 cm^{-1} to 3228 cm^{-1} and 3165 cm^{-1} in FIIR spectra, respectively. The new peaks regarding the stretching vibration of protonated amine groups (-NH_3^+) at around 2823 cm^{-1} and 2596 cm^{-1} can be clearly observed in $\text{CONH-R-NH}_2@\text{PbI}_2+\text{HNO}_3$, indicating that the -NH_2 group was protonated via the H^+ ionized by HNO_3 ³⁸. Meanwhile, the new peak appears at around 2535 cm^{-1} and 1514 cm^{-1} , which are assigned to the stretching vibration of protonated amide (-NH_2^+) and protonated amide II, manifesting the -CONH was protonated by H^+ ^{39,40}. Besides, the peak positions of the stretching vibration of -C=O also obviously shifts from 1550 cm^{-1} to 1606 cm^{-1} . The reason is attributed to that H^+ ionized by HNO_3 coordinates with the lone pair on the N atom on the -CONH , thus breaking the conjugated system of the -CONH- and making the C=O bond shift to high frequency due to its enhanced double bond properties^{41,42}. The above results show that the Pb desorption is achieved based on the protonation of CONH-R-NH_2 molecular configuration by HNO_3 .

Response to the comments of Reviewer #2:

Pb contamination risk is one the most important factors that influence the commercialization of PSCs. Luo et al. were inspired by the spider web and then implanted a multifunctional mesoporous amino-grafted-carbon net into PSCs. The study has indicated that amino-grafted-carbon net can serve as a biomimetic “cage traps” for mitigating Pb leakage. Besides, they have established a sustainable closed-loop Pb management engineering via four critical steps including Pb precipitation, Pb adsorption, Pb desorption, and Pb recycling. It's a fascinating work supported by detailed characterization and data. Even so, there are some problems that need to be clarified before accepting.

Reply: We are grateful to reviewer for the positive comments to improve the manuscript. According to the suggestions, we have made revisions to our manuscript based on the suggestions. All changes have been highlighted in blue in the revised manuscript.

1. The author mentioned that the properties the Yellow River water affect the adsorption capacity of BCT, thus resulting in the increasing of Pb leakage of broken end-of-life devices. However, the Pb leakage rates of BG-PSC and G-PSC in the DI water only are 0.14% and 40.56%, whereas their Pb leakage rates in Yellow River water increase by 8.26% and 20.8%, respectively. Compared with that in DI water, why does the Pb leakage rates increase for BG-PSC while reduce for G-PSC in Yellow River water. Moreover, the data described in the text is inconsistent with the Supplementary Figure 13, please correct it.

Reply: We thank the reviewer for the insightful comments. Firstly, we corrected the data described in the text is inconsistent with the Supplementary Figure 13. And we have revised “However, as shown in Supplementary Fig. 13, it is worth noting that the Pb leakage rates of BG-PSC and G-PSC in the DI water only are 0.14% and 40.56%,”

whereas their Pb leakage rates in Yellow River water increase by 8.26% and **20.8%**, respectively.” as “It is worth noting that the Pb leakage rates of BG-PSC and G-PSC in Yellow River water increase by 8.26% and **20.82%**, respectively.” Then, in this work, as shown in **Figure. 3e, Figure R11 and Table R1**, compared with U-PSC (100%), the Pb leakage rates of G-PSC and BG-PSC in the DI water are only 40.56% and 0.14%, respectively. The Pb leakage rates of G-PSC and BG-PSC in the Yellow River water are 61.38% and 8.40%, respectively. Both G-PSC and BG-PSC have increased Pb leakage rates in Yellow River water compared with that in DI water. Compared with that in DI water, the Pb leakage rates for both the G-PSC and BG-PSC are increased in Yellow River water. Note that the Pb leakage rate of G-PSC increased from 40.56% in DI water to 61.38% in Yellow River water, with an increasing amount of 20.82%. The Pb leakage rate of BG-PSC was 0.14% in DI water and 8.40% in Yellow River water, with an increasing amount of 8.26%. In general, BG-PSC shows lower Pb leakage than G-PSC when changing the scenario from DI water to Yellow River, indicating that BG-PSC exhibits better Pb segregation capacity in both laboratory and real-world scenarios.

Figure R11 (Supplementary Figure. 20 in the revised Supplementary information). Comparison of Pb leakage rates of G-PSC and BG-PSC in deionized water and Yellow River water.

Table R1 Comprehensive comparison and summary of Pb leakage rates of G-PSC and BG-PSC in deionized water and Yellow River water.

Device types	Testing environment	Pb leakage rate (%)	Increasing amount of Pb leakage rate (%)	Pb sequestration efficiency (%)
G-PSC	DI water	40.56	---	59.44
G-PSC	Yellow river	61.38	20.82	38.62
BG-PSC	DI water	0.14	---	99.86

Revisions in the manuscript:

(1) Line 400, Page 14:

It is worth noting that the Pb leakage rates of BG-PSC and G-PSC in Yellow River water increase by 8.26% and 20.82%, respectively.

(2) Line 395-400, Page 14:

However, as shown in Supplementary Fig. 20 and Supplementary table 2, compared with U-PSC (100%), the Pb leakage rates of G-PSC and BG-PSC in the DI water are only 40.56% and 0.14%, respectively. The Pb leakage rates of G-PSC and BG-PSC in the Yellow River water are 61.38% and 8.40%, respectively. It is worth noting that the Pb leakage rates of BG-PSC and G-PSC in Yellow River water increase by 8.26% and 20.82%, respectively.

2. The author proved excellent water vapor absorption characteristics of BCT (Fig. 4f) in Line 357. So, how to understand the synergistic roles in terms of internal secondary water vapor adsorption and external water vapor isolation of the BCT.

Reply: We thank the reviewer for their constructive comment. In this work, the glass substrate with BCT protection layer was encapsulated onto the PSCs via sealing UV glue around the device edge, providing different roles in different scenarios. For example, when the PSCs encapsulated with glass(G-PSC) are broken, the moisture in the humid environment will directly contact with the PSCs as the encapsulated glass substrate is broken, resulting in the rapid degradation of the perovskite light absorption layer, as shown in **Figure R12A**. However, when PSCs encapsulated by BCT(BG-PSC) are broken, the BCT coating layer encapsulated on the device can form a thick water vapor barrier layer on the surface of the device because of its compactness. The barrier layer is able to isolate the water vapor from the perovskite layer when water vapor enters the device through the broken glass, thus isolating external water vapor, as shown in **Figure R12b**. In addition, when the UV glue encapsulated at the edge of G-PSC is aged, water vapor can enter the device through the gap of the aged UV glue, leading to the degradation of the perovskite light-absorbing layer, as shown in **Figure R12c**. However, when aging UV glue cracks appear in BG-PSC, the dense BCT coating layer can secondary adsorb internal water vapor which have entered the inner of the device via the gap of UV glue due to the unique porous structure of BCT, slowing down the erosion of perovskite layer by water vapor (**Figure R12d**). Therefore, the BCT coating layer achieves the synergistic roles of secondary external water vapor isolation and water vapor adsorption based on water vapor isolation and absorption, improving the humidity stability of PSCs.

Figure R12 (Supplementary Figure. 26 in the revised Supplementary information). The synergistic roles in terms of internal secondary water vapor adsorption and external water vapor isolation of the BCT. Comparison of water vapor isolation behavior between glass encapsulated PSCs(a) and BCT encapsulated PSCs(b) after crushing. Comparison of water vapor adsorption behavior between glass encapsulated PSCs(c) and BCT encapsulated PSCs(d) after the degradation of UV glue.

Revisions in the Supporting Information:

Supplementary Figure 26. Comparison of water vapor isolation behavior between glass encapsulated PSCs (a) and BCT encapsulated PSCs (b) after crushing. Comparison of water vapor adsorption behavior between glass encapsulated PSCs (c) and BCT encapsulated PSCs (d) after the degradation of UV glue. In this work, the glass substrate with BCT protection layer was encapsulated onto the PSCs via sealing UV glue around the device edge, providing different roles in different scenarios. For example, when the PSCs encapsulated with glass(G-PSC) are broken, the moisture in the humid environment will directly contact with the PSCs as the encapsulated glass

substrate is broken, resulting in the rapid degradation of the perovskite light absorption layer, as shown in **Supplementary Figure 26a**. However, when PSCs encapsulated by BCT(BG-PSC) are broken, the BCT coating layer encapsulated on the device can form a thick water vapor barrier layer on the surface of the device because of its compactness. The barrier layer is able to isolate the water vapor from the perovskite layer when water vapor enters the device through the broken glass, thus isolating external water vapor, as shown in **Supplementary Figure 26b**. In addition, when the UV glue encapsulated at the edge of G-PSC is aged, water vapor can enter the device through the gap of the aged UV glue, leading to the degradation of the perovskite light-absorbing layer, as shown in **Supplementary Figure 26c**. However, when aging UV glue cracks appear in BG-PSC, the dense BCT coating layer can secondary adsorb internal water vapor which have entered the inner of the device via the gap of UV glue due to the unique porous structure of BCT, slowing down the erosion of perovskite layer by water vapor (**Supplementary Figure 2d**). Therefore, the BCT coating layer achieves the synergistic roles of secondary external water vapor isolation and water vapor adsorption based on water vapor isolation and absorption, improving the humidity stability of PSCs.

3. The purity of recycled PbI_2 is higher than the commercial fresh PbI_2 . Why does the performance of device based on recycled PbI_2 is lower than that based commercial fresh PbI_2 .

Reply: We thank for the reviewer's comment. As shown in **Figure 5e**, there exist a small deviation from stoichiometric ratio of Pb and I element in recycled PbI_2 sample. The recycled PbI_2 has 103% of relative Pb concentration and 100.3% relative iodine concentration compared with commercial PbI_2 (set as a standard control sample with 100% concentration). The actual chemical ratio of I to Pb is 1.93449, indicating the deficiency of the I component. Therefore, the recycled samples should be rectified to $\text{PbI}_{1.93449}$. The excess stoichiometric of Pb may decrease the PCE performance of the devices that fabricated based on recycled PbI_2 due to production of more defect states.

Figure 5e. Relative Pb concentration and I concentration in DMF solution with the equivalent amount of commercial fresh 99.99% PbI_2 and recycled PbI_2 , where the commercial fresh 99.99% PbI_2 was used as the reference. Pb and I concentration were determined via ICP-OES.

Revisions in the manuscript:

Line 536-537, Page 19 and Line 539-543, Page 19:

As shown in Fig. 5e, there exist a small deviation from stoichiometric ratio of Pb and I element in recycled PbI_2 sample. The recycled PbI_2 has 103% of relative Pb concentration and 100.3% relative iodine concentration compared with commercial PbI_2 (set as a standard control sample with 100% concentration). The actual chemical ratio of I to Pb is 1.93449, indicating the deficiency of the I component. Therefore, the recycled samples should be rectified to $\text{PbI}_{1.93449}$. The excess stoichiometric of Pb may decrease the PCE performance of the devices that fabricated based on recycled PbI_2 due to production of more defect states.

4. The intensity of XPS data in this paper needs to be normalized.

Reply: We thank the reviewer for the constructive suggestion. We have normalized the intensity of the XPS data in the manuscript, the modified **Supplementary Figure 3 (Figure R13)** is as follows.

Figure R13 (Supplementary Figure. 3 in the revised Supplementary information). XPS spectra of Pb4f for MM, BCT and PbI_2 @BCT.

Response to the comments of Reviewer #3:

This work describes Pb capturing idea for sustainable perovskite solar cells (PSCs). Bio-inspired cage traps for the Pb ions were designed and its function of immobilization of Pb ions was studied using acidic water. In addition, bio-inspired system was applied to PSCs, which was studied in terms of Pb immobilization ability and photovoltaic performance. This work is recommended for publication after minor revision.

Reply: We thank the reviewer for the positive comments and valuable suggestions of our manuscript. We have addressed all the comments from the reviewer in the revised the manuscript. All changes have been highlighted in blue in the revised manuscript.

1. What about Pb immobilization efficiency at high pH aqueous condition?

Reply: We thank reviewer for helpful suggestion. According to the suggestion, we have

conducted additional experiments to provide the Pb immobilization efficiency of unpackaged perovskite solar cell (U-PSC), glass encapsulated perovskite solar cell (G-PSC) and BCT encapsulated PSC (BG-PSC) at aqueous condition with pH=10 and pH=11, respectively (**Figure R14**). As shown in **Figure R14a**, the leaked Pb ions for U-PSC and G-PSC are 7.82 and 2.58 ppm respectively, at pH=10 aqueous condition. In contrast, only 0.523 ppm of Pb ions leaked from BG-PSC, achieving 93.31% of the Pb sequestration efficiency. Meanwhile, in pH=11 aqueous condition, compared with the Pb leakage concentration of 6.27 ppm for U-PSC, the Pb leakage concentration of BG-PSC is only 0.461 ppm, achieving 92.64% of the Pb sequestration efficiency, whereas G-PSCs have Pb leakage concentration and Pb leakage rate as high as 2.17 ppm and 34.61%, respectively (**Figure R14b**). It can be noted that the BCT package can still exhibit over 90% Pb immobilization efficiency even in high pH aqueous conditions.

Figure R14 (Supplementary Figure. 18 in the revised Supplementary information). The Pb immobilization efficiency of U-PSC, G-PSC and BG-PSC at aqueous condition with pH=10(a) and pH=11(b).

Revisions in the manuscript:

Line 373-380, Page 13 and Line 381-384, Page 14:

Moreover, we performed experiments to verify the Pb immobilization efficiency of cracked U-PSC, G-PSC and BG-PSC in an aqueous condition with pH of 10 and 11, respectively (Supplementary Fig. 18). As shown in Supplementary Fig. 18a, the leaked Pb ion concentrations for U-PSC and G-PSC are 7.82 and 2.58 ppm, respectively, at pH=10 aqueous condition. In contrast, only 0.523 ppm of Pb ions leaked from BG-PSC, achieving 93.31% of the Pb sequestration efficiency. Meanwhile, in pH of 11 aqueous condition, compared with the Pb leakage concentration of 6.27 ppm for U-PSC, the Pb leakage concentration of BG-PSC was only 0.461 ppm, achieving a Pb sequestration efficiency of 92.64%, whereas G-PSCs had Pb leakage concentration and Pb leakage rate as high as 2.17 ppm and 34.61%, respectively (Supplementary Fig. 18b). It is noted that the BCT package can still exhibit over 90% Pb immobilization efficiency even in high pH aqueous conditions.

2. What about PV performance under 85% RH and 85 °C?

Reply: We thank the reviewer for the constructive comments. According to the suggestion, we have performed additional accelerated aging tests complying with

international summit on organic photovoltaic stability (ISOS) standards to evaluate the PV performance and stability of unpackaged PSCs (U-PSCs), glass encapsulated PSCs (G-PSCs) and BCT encapsulated PSCs (BG-PSCs), respectively. All the tested devices were integrated with a configuration of indium tin oxide (ITO)/tin oxide (SnO_2)/FA-based perovskite/Spiro-OMeTAD/gold (Au). Firstly, the U-PSC, G-PSC and BG-PSC were exposed to a moisture environment with 85% RH for over 600 h at ambient temperature (25°C) and the evolution of PCEs were tracked to figure out the effect of the humidity on device degradation. As shown in **Figure R15a**, the PCEs of the U-PSC and the G-PSC drop to 47.92% and 79.22% of their initial PCEs, respectively. It is worth noting that the BG-PSCs exhibit promising stability, maintain 98.40% of their initial PCE, due to the synergistic role of the BCT in terms of internal secondary water vapor adsorption and external water vapor isolation. Secondly, the thermal stability of the prepared U-PSCs, G-PSCs and BG-PSCs with device structure of ITO/ SnO_2 /perovskite/Spiro-OMeTAD/Au under 85°C temperature in N_2 atmosphere was tested. However, the PCEs of U-PSCs, G-PSCs and BG-PSCs dropped dramatically to 6.87%, 11.62% and 60.87% of their initial PCEs after 24 h of heat accelerated aging. The reasons for the rapid thermal degradation of devices can be ascribed to the following reasons: (i) the decomposition of perovskite crystalline structure, evolving from tetragonal phase to trigonal PbI_2 layered crystals at 85°C , resulting in a failure of light absorption characteristic [*Joule* **1**, 548(2017).]; (ii) Spiro-MeOTAD with lithium bis(trifluoromethanesulfonyl) imide (LiTFSI) and tert-butylpyridine (TBP) we used was crystallized at 85°C due to a low glass transition temperature, and hole mobility was significantly deteriorated, which was responsible for the thermal instability. [*ACS Appl. Mater. Interfaces* **9**, 7148–7153(2017)]; (iii) gold (Au) migrated into perovskite material through Spiro-MeOTAD, which in turn seriously affects the device performance under working conditions [*ACS Nano* **10**, 6306–6314(2016)]. Please note that although some reported works have achieved good performance of thermal stability, the reason is mainly ascribed to that they replaced doped Spiro-OMeTAD with undoped PTAA as or only heated the optimized perovskite film without coating hole transport layer (HTL) and electrode. Here, we devoted to the evaluation of thermal stability based on the complete device configurations with raw materials, which would be more reliable and objective. Challenges still remain to ameliorate the thermal tolerance limit of the PSCs. Thirdly, we additionally tested the light stability of U-PSCs, G-PSCs and BG-PSCs under continuous illumination of AM1.5G, 100 mW/cm^2 with mild environment temperature of $60\pm 5^\circ\text{C}$. As shown in **Figure R15b**, it is noted that the BG-PSC is able to maintain its initial PCE of 81.49%, whereas the PCE of U-PSCs and G-PSCs drop dramatically to 43.65% and 36.62%, respectively. This indicates that dark BCT materials with light adsorption property may be able to partially shield against damage from thermal radiation originating from the secondary reflection of light from the side of gold electrode during continuous light exposure. Based on the above discussion, it can be concluded that BCT packaging technology can enhance the operational stability of PSCs under extreme environmental conditions, which is conducive to the commercialization of PSCs.

Figure R15. a, Normalized PCE of U-PSC, G-PSC and BG-PSC stored in a moisture environment with 85% RH for over 600h at 25°C (**Figure 4g in the revised manuscript**). b, Normalized PCE of U-PSC, G-PSC and BG-PSC under continuous light at AM1.5G, 100 mW/cm² with 60±5°C. (**Figure 4e in the revised manuscript**).

Revisions in the manuscript:

Line 444-454, Page 16 and Line 460-466, Page 16:

In order to evaluate the stability of PSCs under international summit on organic photovoltaic stability (ISOS) standards, the stabilities of U-PSC, G-PSC and BG-PSC were performed under continuous light at AM1.5G, 100 mW/cm² with 60±5°C, 50% relative humidity at 25°C and 85% relative humidity at 25°C, respectively (Supplementary Fig. 25). As shown in Fig. 4e, it was noted that the BG-PSC was able to maintain its initial PCE of 81.49% under AM1.5G, 100 mW/cm² at 60±5°C over 360 h, whereas the PCE of U-PSCs and G-PSCs dropped dramatically to 43.65% and 36.62%, respectively. This indicated that dark BCT materials with light adsorption property may be able to partially shield against damage from thermal radiation originating from the secondary reflection of light from the side of gold electrode during continuous light exposure. When the U-PSC, G-PSC and BG-PSC were exposed to a moisture environment with 50% RH for over 1000h at 25°C, the PCE for U-PSC and G-PSC decreased dramatically to 56% and 62% of the initial PCE after the 1000-hour test respectively, respectively, which can be attributed to the degradation of the perovskite layer due to the moisture entering into the device through the ultraviolet (UV)

curing adhesive gap. The PCE of BG-PSC could retain 92% of the initial PCE (Fig. 4f). As shown in Fig. 4g, the PCEs of the U-PSC and the G-PSC dropped to 47.92% and 79.22% of their initial PCEs at a moisture environment with 85% RH for over 600h at 25°C, respectively. It is worth noting that the BG-PSCs exhibit promising stability and maintained 98.40% of their initial PCE, due to the synergistic role of the BCT in terms of internal secondary water vapor adsorption and external water vapor isolation, preventing the erosion impact of moisture on perovskite material (Fig. 4a and Supplementary Fig. 26).

Response to the comments of Reviewer #4:

This work reported a novel bioinspired sealant material, entitled "cage traps" (BCT), that takes inspiration from spider webs. This encapsulation strategy will protect the devices from the erosion of humidity and act as a blocker to adsorb Pb that might leak. The working mechanism of the BCT strategy involves a synergistic approach that combines chemical chelation and physical adsorption to address Pb toxicity and the instability of perovskite modules. The author also highlights the simulated real-world test conditions by assessing the Pb contamination in Yellow River and surrounding soils. This research provides valuable insights into material design and prudent tests, which sets the groundwork for further research despite some controversial points. Hence, the work warrants publication in Nature Communication after addressing the following issues.

Reply: We would like to thank the reviewer very much for the positive evaluation of our manuscript. All the comments are very helpful for improving our work. We have addressed all the comments and revised the manuscript carefully. All changes have been **highlighted in blue** in the revised manuscript.

1. Since the photovoltaic performance is mentioned repeatedly in Fig. 4b and Fig. 5f, there appears to be a noticeable difference in the short-circuit current density (J_{sc}), which decreases from approximately 25.8 to 24.8 mA/cm². This discrepancy has led to some confusion regarding the possible causes of this J_{sc} loss, whether it could be attributed to the encapsulation process or if the performance analysis of the devices lacks rigor. Please explain the J_{sc} difference between the two figures and improve the self-consistency of the analysis.

Reply: We would like to thank the reviewer for the professional comment. In this work, we prepared two batches of devices in March and October, and encapsulated with BCT layers to verify the compatibility of BCT with the photovoltaic performance of PSCs. The box line statistics of V_{oc} , PCE, FF and J_{sc} before and after the device encapsulating are shown in **Figure R16**. The statistics of V_{oc} , PCE, FF and J_{sc} values before and after packaging are shown in the **Table R2** and **R3**. It can be seen that the J_{sc} variation range of devices before and after encapsulation is 23.34844-25.40793 mA·cm² and 23.25188-25.72475 mA·cm², with the average J_{sc} of 24.57167 mA·cm² and 24.24293 mA·cm², respectively. It shows that encapsulating the BCT layer does not cause J_{sc} loss of the device. Furthermore, the average J_{sc} of the devices prepared in October is 25.14652

$\text{mA}\cdot\text{cm}^2$ which is slightly higher the average J_{SC} of $24.06069 \text{ mA}\cdot\text{cm}^2$ for the devices prepared in March. The reason is attributed to the seasonal difference between March and October caused significant changes in climate temperature, humidity of experimental operating environment and glove box temperature during the preparation of PSCs by two-step method. The crystalline quality of perovskite thin films, such as grain size and density of defect states, is affected, thus influencing device performance [J. Mater. Chem. A **3**, 19901-19906 (2015). Nat. Commun. **13**, 4891(2022)]. Therefore, the J_{SC} of the devices prepared by us at the end of September based on the recovered PbI_2 is higher than that prepared in March.

Figure R16. Box drawings of V_{OC} (a), PCE(b), FF(c) and J_{SC} (d) of PSCs before and after BCT packaging.

Table R2. Summary table of V_{OC} , PCE, FF and J_{SC} of PSCs prepared in March before and after BCT packaging.

Before encapsulation				After encapsulation			
V_{oc} (V)	PCE (%)	FF (%)	J_{sc} ($\text{mA}\cdot\text{cm}^2$)	V_{oc} (V)	PCE (%)	FF (%)	J_{sc} ($\text{mA}\cdot\text{cm}^2$)
1.14821	21.11825	76.34076	24.05412	1.11149	20.24374	75.78803	24.13488
1.13404	19.42361	70.60051	24.22154	1.13754	20.8218	75.52234	24.34093
1.16214	20.50761	73.32258	24.0285	1.16686	19.43292	70.8615	23.60316
1.14402	20.23848	73.6585	23.97883	1.12484	20.52186	76.63305	23.90951
1.08084	19.98742	78.62717	23.48179	1.13503	19.68643	74.91395	23.25188
1.12827	18.78297	71.18725	23.34844	1.14895	19.69767	73.55286	23.40853
1.11665	21.52715	78.01087	24.7035	1.125	21.22465	77.08864	24.47175
1.11998	22.11932	79.46859	24.8434	1.13411	22.20421	79.58831	24.59789
1.17036	20.58457	73.32704	23.88616	1.16621	20.64828	75.84013	23.40202

Table R3. Summary table of V_{OC} , PCE, FF and J_{SC} of PSCs prepared in October before and after BCT packaging.

Before encapsulation				After encapsulation			
V_{oc} (V)	PCE (%)	FF (%)	J_{sc} (mA·cm ²)	V_{oc} (V)	PCE (%)	FF (%)	J_{sc} (mA·cm ²)
1.132	21.20977	73.87327	25.354	1.156	21.90729	73.66285	25.72475
1.13195	21.0465	73.71229	25.21483	1.15274	22.43101	76.00682	25.59957
1.10656	21.7649	78.36026	25.09167	1.11119	21.13244	77.65584	24.48808
1.11038	22.4877	79.68	25.40793	1.12001	22.03548	79.40701	24.77479
1.14912	22.29013	77.2358	25.10582	1.1499	21.80619	78.47448	24.16353
1.12725	19.64412	70.22923	24.80507	1.15412	20.25686	74.42843	23.58042
1.13836	21.27692	74.8533	24.961	1.12425	21.22087	77.80947	24.25695
1.11974	22.56927	79.8541	25.23183	1.11274	21.25825	78.22323	24.42114

Revisions in the manuscript:

Line 544-554, Page 19:

It shows that FA-based PSCs prepared by recycled PbI₂ display 22.08% PCE with an open circuit voltage (V_{oc}) of 1.11 V, a short-circuit current density (J_{sc}) of 25.87 mA/cm², a fill factor (FF) of 76.89%, which is almost close to the device fabricated on commercial fresh PbI₂ achieves a PCE of 22.37% with a V_{oc} of 1.13 V, a J_{sc} of 25.66 mA/cm², and a FF of 77.03% (Fig. 5f). Please note that the device of J_{sc} here higher than that of U-PSC and BG-PSC in Fig. 4b is attributed to the seasonal difference between March and October caused significant changes in climate temperature, humidity of experimental operating environment and glove box temperature during the preparation of PSCs by two-step method. The crystalline quality of perovskite thin films, such as grain size and density of defect states, is affected, thus influencing device performance^{50,51}.

2. The commercialization of perovskite solar cells is required to withstand moisture ingress and cyclic temperature extremes. In light of these requirements, recent studies have explored the ambient and operational stability (MPP stability) of different encapsulation and adsorption materials in enhancing the stability of perovskite solar cells with more rigorous criteria. (Sci. Adv. 2021, 7: eabi8249; Nat. Energy 2020, 5, 1003–1011) I recommend the author add some stability evaluation from different angles.

Reply: We would like to thank the reviewer for the constructive comments. According to the suggestion, we have performed additional accelerated aging tests complying with international summit on organic photovoltaic stability (ISOS) standards to evaluate the PV performance and stability of unpackaged PSCs (U-PSCs), glass encapsulated PSCs (G-PSCs) and BCT encapsulated PSCs (BG-PSCs), respectively. All the tested devices were integrated with a configuration of indium tin oxide (ITO)/tin oxide (SnO₂)/FA-based perovskite/Spiro-OMeTAD/gold (Au). Firstly, the U-PSC, G-PSC and BG-PSC were exposed to a moisture environment with 85% RH for over 600 h at ambient temperature (25°C) and the evolution of PCEs were tracked to figure out the effect of the humidity on device degradation. As shown in **Figure R15a**, the PCEs of the U-PSC and the G-PSC drop to 47.92% and 79.22% of their initial PCEs, respectively. It is worth nothing that the BG-PSCs exhibit promising stability, maintain 98.40% of their

initial PCE, due to the synergistic role of the BCT in terms of internal secondary water vapor adsorption and external water vapor isolation. Secondly, we additionally tested the light stability of U-PSCs, G-PSCs and BG-PSCs under continuous illumination of AM1.5G, 100 mW/cm² with mild environment temperature of 60±5°C. As shown in **Figure R15b**, it is noted that the BG-PSC is able to maintain its initial PCE of 81.49%, whereas the PCE of U-PSCs and G-PSCs drop dramatically to 43.65% and 36.62%, respectively. This indicates that dark BCT materials with light adsorption property may be able to partially shield against damage from thermal radiation originating from the secondary reflection of light from the side of gold electrode during continuous light exposure. Based on the above discussion, it can be concluded that BCT packaging technology can enhance the operational stability of PSCs under extreme environmental conditions, which is conducive to the commercialization of PSCs.

Figure R17. a, Normalized PCE of U-PSC, G-PSC and BG-PSC stored in a moisture environment with 85% RH for over 600h at 25°C (**Figure 4g in the revised manuscript**). b, Normalized PCE of U-PSC, G-PSC and BG-PSC under continuous light at AM1.5G, 100 mW/cm² with 60±5°C (**Figure 4e in the revised manuscript**).

Revisions in the manuscript:

Line 444-454, Page 16 and Line 460-466, Page 16:

In order to evaluate the stability of PSCs under international summit on organic photovoltaic stability (ISOS) standards, the stabilities of U-PSC, G-PSC and BG-PSC were performed under continuous light at AM1.5G, 100 mW/cm² with 60±5°C, 50%

relative humidity at 25°C and 85% relative humidity at 25°C, respectively (Supplementary Fig. 25). As shown in Fig. 4e, it was noted that the BG-PSC was able to maintain its initial PCE of 81.49% under AM1.5G, 100 mW/cm² at 60±5°C over 360 h, whereas the PCE of U-PSCs and G-PSCs dropped dramatically to 43.65% and 36.62%, respectively. This indicated that dark BCT materials with light adsorption property may be able to partially shield against damage from thermal radiation originating from the secondary reflection of light from the side of gold electrode during continuous light exposure. When the U-PSC, G-PSC and BG-PSC were exposed to a moisture environment with 50% RH for over 1000h at 25°C, the PCE for U-PSC and G-PSC decreased dramatically to 56% and 62% of the initial PCE after the 1000-hour test respectively, respectively, which can be attributed to the degradation of the perovskite layer due to the moisture entering into the device through the ultraviolet (UV) curing adhesive gap. The PCE of BG-PSC could retain 92% of the initial PCE (Fig. 4f). As shown in Fig. 4g, the PCEs of the U-PSC and the G-PSC dropped to 47.92% and 79.22% of their initial PCEs at a moisture environment with 85% RH for over 600h at 25°C, respectively. It is worth noting that the BG-PSCs exhibit promising stability and maintained 98.40% of their initial PCE, due to the synergistic role of the BCT in terms of internal secondary water vapor adsorption and external water vapor isolation, preventing the erosion impact of moisture on perovskite material (Fig. 4a and Supplementary Fig. 26).

3. In Fig. 4e of the manuscript, the author evaluated the stability of encapsulated PSCs with a performance test every 250 hours for the entire duration of 1000 hours. It seems abnormal as compared to other reported works aimed at preventing Pb leakage, where the stability test is usually conducted in less than every 100 hours for each data point. (Sci. Adv. 2021, 7: eabi8249; ACS Energy Lett. 2021, 6, 3443–3449; Adv. Funct. Mater. 2022, 32, 2201036) It's recommended to narrow the interval time between data points of the performance evolution to obtain a more robust and detailed result of the stability of encapsulated PSCs.

Reply: We would like to thank the reviewer for the insightful comments. We have increased the interval time between data points of the performance evolution in the Figure 4e to obtain a more robust and detailed result of the stability of encapsulated PSCs. The modified Figure 4e (**Figure R18**) is as follows.

Figure R18 (Figure 4f in the revised manuscript). Normalized PCE of U-PSC, G-PSC and BG-PSC stored in a moisture environment with 50% RH for over 1000h at 25°C.

4. While the author of this work demonstrated back encapsulation of BCT, several other works highlight the importance of encapsulation from both the front and back sides, as indicated in Supplementary Table 1. The question arises whether there is still a possibility of Pb leakage from the front side and cross-sections in extreme scenarios when the devices are cracked into pieces. Therefore, it would be worthwhile to investigate the robustness of the encapsulation technique under such conditions to comprehensively evaluate its efficacy.

Reply: We would like to thank the reviewer for the valuable comments. Most of the published works in Supplementary Table 1 focus on developing semi-transparent Pb adsorbent, which can be installed on the light incident and light- accepting side of PSCs. In order to diminish photovoltaic performance loss, high optical transparency is a necessary prerequisite for allowing adequate light quantity to pass through. Therefore, the thickness of semi-transparent Pb adsorbent needs to be strictly limited, which severely compromises the Pb adsorption capacity and ambient resistance. In the actual situation, the leaked Pb components are inclined to flow to the backlight side due to the influence of gravity, whereas the corresponding materials or strategy is still limited, as shown in **Figure R19**. Therefore, in this work, we focused on encapsulating environmentally friendly and low-cost Pb capture material layers on the backlight side of PSCs, and in this case, material transparency isn't necessary to be taken into consideration. Benefiting from the unlimited thickness requirement, the BCT exhibits superior Pb capture performance and a steric shield effect under ambient stimulus.

Schematic diagram of lead leakage in practical application of perovskite solar cells

Figure R19 (Supplementary Figure. 13 in the revised Supplementary information). The Pb leakage process of glass-encapsulated PSCs in actual operation.

Besides, according to comments, we conducted additional experiments to evaluate the effect of different install position of functional materials on Pb leakage suppression performance under extreme destructive conditions. First, the mature acrylate ultraviolet curing resin was selected to package on light- accepting sides of PSCs, which have been

demonstrated with capability of Pb leakage suppression in the previous works, and the BCT was installed on the back side of PSCs (BG-PSC) for comparison (**Figure R20a**). The unpackaged perovskite solar cell (U-PSC), glass encapsulated perovskite solar cell (G-PSC), resin encapsulated PSC (R-PSC) and BG-PSC were completely cracked via violent hail impact simulation (**Figure R20b**). Then, we immersed them in DI water for 6h and access the concentration of leaked Pb ions. As shown in **Figure R21**, the leaked Pb ions are 7.84, 5.57 and 4.03 ppm for U-PSC, G-PSC and R-PSC, respectively. Encouragingly, only 0.876 ppm of Pb ions leak from BG-PSC. Compared with U-PSC, G-PSC and R-PSC, which has the Pb sequestration efficiency of only 0.00%, 26.79% and 48.59%, respectively, BG-PSC exhibits Pb sequestration efficiency as high as 88.83%. Therefore, it is concluded that BCT encapsulation strategy displays high Pb sequestration efficiency even when the devices are cracked into pieces, while the resin encapsulation only prevents a small amount of Pb ion leakage. This result demonstrated that the packing the Pb adsorption materials on the backside of the panel is highly demanded when the PSCs are suffered from undesired broken. Besides, the significance of this work is to propose a BCT packaging strategy based on environmentally friendly and low cost BCT materials. The thickness of such material would not be limited by the rules of optical transmission, thus ensuring a superior Pb adsorption capacity and Pb sequestration efficiency. Moreover, the Pb ions captured by the BCT layer can be recycled and reused via the Pb closed-loop management strategy, which significantly increases the environmental friendliness of PSCs and reduces the cost of commercial application of PSCs. In practical applications, BCT materials can be an alternative to cooperate with translucent Pb capture materials for double-sided application to achieve synergistic encapsulation for enhanced Pb leakage suppression.

Figure R20 (Supplementary Figure. 14 in the revised Supplementary information).
a, The methods of encapsulating PSCs with resin and BCT. b, Typical photographs of

completely cracked U-PSC, G-PSC, R-PSC and BG-PSC.

Figure R21 (Supplementary Figure. 15 in the revised Supplementary information). The Pb leakage and Pb sequestration efficiencies of the completely cracked U-PSC, G-PSC, R-PSC and BG-PSC in DI water.

Revisions in the manuscript:

Line 340-353, Page 12 and Line 354-363, Page 13:

When PSCs are subjected to external impacts such as hail during practical application, the leaked Pb components also tend to flow to the backside along the cracks within the damaged PSCs by gravity due to the modules being installed at a large inclination angle to optimize sunlight exposure for that location (Supplementary Fig. 13). Besides, we conducted additional experiments to compare the effect of different installation positions of functional materials on Pb leakage suppression performance under extreme destructive conditions. First, the mature acrylate ultraviolet curing resin was selected to package on light- accepting sides of PSCs, which have demonstrated with excellent capability of Pb leakage suppression in the previous works, and the BCT was installed on the backside of PSCs (BG-PSC) for comparison (Supplementary Fig. 14a). The unpackaged perovskite solar cell (U-PSC), glass-encapsulated perovskite solar cell (G-PSC), resin encapsulated PSC (R-PSC) and BG-PSC were completely cracked via violent hail attacking simulation (Supplementary Fig. 14b). Then, we immersed them in DI water for six hours and measured the concentration of leaked Pb ions. As shown in Supplementary Fig. 15, the leaked Pb ion concentrations were 7.84, 5.57 and 4.03 ppm for U-PSC, G-PSC and R-PSC, respectively. Encouragingly, only

0.876 ppm of Pb ions leaked from BG-PSC. Compared with U-PSC, G-PSC and R-PSC, which exhibit the Pb sequestration efficiency of only 0.00%, 26.79% and 48.59%, respectively, BG-PSC exhibits Pb sequestration efficiency as high as 88.83%. Therefore, it is concluded that the BCT encapsulation strategy displays much higher Pb sequestration efficiency compared with the conventional encapsulation strategy of packing the material on light-accepting glass side, even when the devices are cracked into pieces (Supplementary Fig. 16). This is mainly attributed to the advantage of large adsorption capacity and fast adsorption rate for BCT materials.

5. Suggest improvements. Fig. 1a is not friendly to readers with Arachnophobia, we suggest the author replace the schematic illustration with only spider webs.

Reply: We would like to thank the reviewer for the comments. We have modified Figure 1a based on your suggestion, as shown in **Figure R22**.

Figure R22 (Figure 1a in the revised manuscript). Schematic of spider ensnare prey (left) and Pb chelation behavior of BCT (right).

6. Compared with some stretchable and self-healable polymer materials for the sequestration of Pb, are there any advantages of the BCT materials for further commercialization of PSCs?

Reply: We are grateful to the reviewer for the comments. As shown in **Figure R23**, our study focuses on the Pb sequestration of perovskite solar cells (PSCs) through the use of BCT materials as a sequestration agent. While there are stretchable and self-healable polymer materials available for this purpose, they have some drawbacks that may impact their suitability for large-scale commercialization.

Polymer materials offer rapid chemical Pb capture, suppress Pb leakage in PSCs, and improve their overall stability. However, they suffer from several significant disadvantages. Firstly, most organic polymer materials have high preparation costs, making them unsuitable for large-scale industrial usage. Secondly, these polymer materials often lack environmental tolerance and may fail due to factors such as lighting, high temperatures, and humidity when packaged behind the ITO/FTO substrate. This

can lead to a loss of the Pb leakage inhibiting effect of PSCs. Thirdly, some polymer materials have secondary pollution characteristics that negatively impact the environmentally friendly profile of PSCs [*Trends Chem.* **1**, 148-151(2019)].

In contrast, BCT materials offer several advantages over polymer materials. Firstly, they have a low preparation cost, making them suitable for large-scale industrial usage, which reduces the overall cost of commercializing PSCs. Secondly, BCT materials have stable environmental tolerance, making them less susceptible to failure due to environmental factors such as lighting, high temperature, and humidity. Thirdly, BCT materials are environmentally friendly and do not have secondary pollution characteristics, unlike some polymer materials. Finally, BCT materials can recycle Pb in waste PSCs, which promotes the green commercialization of PSCs.

Overall, BCT materials offer an efficient way to reduce Pb contamination during the actual operating process of PSCs while reducing the cost of PSC production. They also offer a more environmentally friendly alternative to polymer materials and can promote the green commercialization of PSCs. Therefore, we believe that BCT materials are a promising alternative to polymer materials for sequestration of Pb in PSCs.

Thank you again for your valuable comments, and we hope that this revised response meets your satisfaction.

Figure R23. (Supplementary Figure. 31 in the revised Supplementary information). The comprehensive comparison of polymeric material encapsulation and BCT encapsulation for reducing Pb contamination levels in PSCs.

Revisions in the Supporting Information:

Supplementary Figure 31. The comprehensive comparison of polymeric material encapsulation and BCT encapsulation for reducing Pb contamination levels in PSCs. As shown in **Supplementary Figure 31**, our study focuses on the Pb sequestration of perovskite solar cells (PSCs) through the use of BCT materials as a sequestration agent. While there are stretchable and self-healable polymer materials available for this purpose, they have some drawbacks that may impact their suitability for large-scale commercialization.

Polymer materials offer rapid chemical Pb capture, suppress Pb leakage in PSCs, and improve their overall stability. However, they suffer from several significant disadvantages. Firstly, most organic polymer materials have high preparation costs, making them unsuitable for large-scale industrial usage. Secondly, these polymer materials often lack environmental tolerance and may fail due to factors such as lighting, high temperatures, and humidity when packaged behind the ITO/FTO substrate. This can lead to a loss of the Pb leakage inhibiting effect of PSCs. Thirdly, some polymer materials have secondary pollution characteristics that negatively impact the environmentally friendly profile of PSCs³.

In contrast, BCT materials offer several advantages over polymer materials. Firstly, they have a low preparation cost, making them suitable for large-scale industrial usage, which reduces the overall cost of commercializing PSCs. Secondly, BCT materials have stable environmental tolerance, making them less susceptible to failure due to environmental factors such as lighting, high temperature, and humidity. Thirdly, BCT materials are environmentally friendly and do not have secondary pollution characteristics, unlike some polymer materials. Finally, BCT materials can recycle Pb in waste PSCs, which promotes the green commercialization of PSCs.

Overall, BCT materials offer an efficient way to reduce Pb contamination during the actual operating process of PSCs while reducing the cost of PSC production. They also offer a more environmentally friendly alternative to polymer materials and can promote the green commercialization of PSCs. Therefore, we believe that BCT materials are a promising alternative to polymer materials for sequestration of Pb in PSCs.

List of changes:

1. We have discussed the new respects of the bioinspired “cage traps” (BCT) from the novelty functional Pb capture molecular structure (CONH-R-NH₂), the new Pb adsorption enhancement mechanism and multifunctional characteristics in section **Introduction** (Line 69-75, Page 3, Line 77-83, Page 4, and Line 93-98, Page 4 in the revised manuscript).
2. We have discussed the limitations of translucent Pb adsorption materials in section **Introduction** (Line 47-53, Page 3 in the revised manuscript).
3. The detailed synthetic material method labeled with bioinspired has been added in section **Chemical synthesis design of the BCT** (Line 102-108, Page 4 and Line 109-115, Page 5 in the revised manuscript).
4. The detailed capillary adsorption action mechanism has been discussed in section **Pb capturing mechanism based on the chemical chelation** (Line 224-229, Page 8 and Line 230-242, Page 9 in the revised manuscript).
5. The specific and in-depth surface area enhancement in the synthesis of BCT has been added in section **Pb capturing mechanism based on the chemical chelation** (Line 194-202, Page 7 and Line 203-212, Page 8 in the revised manuscript).
6. The detailed and in-depth generation mechanism of BCT materials and Pb adsorption mechanism based on FT-IR, XPS, etc. have been discussed in section **Chemical synthesis design of the BCT** and **Pb capturing mechanism based on the chemical chelation** (Line 135-136, Page 5, Line 138-141, Page 5, Line 142-160, Page 6, Line 172-174, Page 6, Line 180-186, Page 7 and Line 189-192, Page 7 in the revised manuscript).
7. The effect of different install position of functional materials on Pb leakage suppression performance under extreme destructive conditions when devices were damaged completely has been added in section **Pb leakage assessment of solar panels under real-world scenarios** (Line 340-353, Page 12 and Line 354-363, Page 13 in the revised manuscript).
8. The Pb release process due to dominated by secondary protonation reaction has been in-depth discussed in section **Pb recycling engineering under sustainable closed-loop management** (Line 479-500, Page 17 and Line 501-513, Page 18 in the revised manuscript).
9. We have corrected the data described in the text is inconsistent with the Supplementary Figure 13 and described the difference of Pb leakage rates of U-PSC and BG-PSC between DI water and Yellow River water (Line 395-400, Page 14 in the revised manuscript).
10. The synergistic roles in terms of internal secondary water vapor adsorption and external water vapor isolation of the BCT was discussed in **Supplementary Figure 25**.
11. The stoichiometric ratio of Pb and I element in recycled PbI₂ was discussed in section **Pb recycling engineering under sustainable closed-loop management** (Line 536-537, Page 19 and Line 539-543, Page 19 in the revised manuscript).
12. We have normalized the intensity of the XPS data in the **Supplementary Figure 3**.

13. The Pb immobilization efficiency of unpackaged perovskite solar cell (U-PSC), glass encapsulated perovskite solar cell (G-PSC) and BCT encapsulated PSC (BG-PSC) at aqueous condition with pH=10 and pH=11 have been added in section **Pb leakage assessment of solar panels under real-world scenarios** (Line 373-380, Page 13 and Line 381-384, Page 14 in the revised manuscript).
14. The stabilities of U-PSCs, G-PSCs and BG-PSCs under continuous light at AM1.5G, 100 mW/cm² with 60±5°C and 85% relative humidity at 25°C have been added in section **The compatibility validation of BCT with the photovoltaic performance and long-term device stability** (Line 444-454, Page 16 and Line 460-466, Page 16 in the revised manuscript).
15. We have explained the J_{SC} difference between the in Fig. 4b and Fig. 5f (Line 544-554, Page 19 in the revised manuscript).
16. We have increased the interval time between data points of the performance evolution in the **Fig. 4e** to obtain a more robust and detailed result of the stability of encapsulated PSCs.
17. We have replaced the spider in **Fig. 1a** with mosquitoes.
18. We have discussed the advantages of the BCT materials for further commercialization of PSCs in the **Supplementary Figure 31**.
19. We have added the $J-V$ characteristics of U-PSC and BG-PSC under reverse or forward scanning direction in **Supplementary Figure 24**.

REVIEWERS' COMMENTS

Reviewer #1 (Remarks to the Author):

I have reviewed your thoughtful response.

There were parts of the text where the novelty and oversimplified discussion of the experimental results obscured the meaning, which made it difficult for me to approve the paper for publication.

However, when I reviewed the manuscript based on the detailed discussion you added, the revised manuscript addressed most of my initial concerns.

Therefore, I believe that the current manuscript is good enough to be published in Nature Communications.

Reviewer #2 (Remarks to the Author):

The author has answered the reviewer's questions well and we agreed to receive the manuscript.

Reviewer #4 (Remarks to the Author):

The quality of the revised manuscript has been greatly improved. I thus recommend to publish it.